# Towards Pre-trained Graph Condensation via Optimal Transport

**Yeyu Yan**[1,2], **Shuai Zheng**[1,2], **Wenjun Hui**[1,2], **Xiangkai Zhu**[3], **Dong Chen**[1,2],
**Zhenfeng Zhu**[1,2*], **Yao Zhao**[1,2], **Kunlun He**[4]

[1]Institute of Information Science, Beijing Jiaotong University, China
[2]Visual Intelligence + X International Cooperation Joint Laboratory of the Ministry of Education, China
[3]Jinan University, China   [4]Chinese PLA General Hospital, China
`yanyeyu-work@foxmail.com, zs1997@bjtu.edu.cn,`
`22110090@bjtu.edu.cn, 18063597830@163.com, dchen2001@bjtu.edu.cn,`
`zhfzhu@bjtu.edu.cn, yzhao@bjtu.edu.cn, kunlunhe@plagh.org`

## Abstract

Graph condensation (GC) aims to distill the original graph into a small-scale graph, mitigating redundancy and accelerating GNN training. However, conventional GC approaches heavily rely on rigid GNNs and task-specific supervision. Such a dependency severely restricts their reusability and generalization across various tasks and architectures. In this work, we revisit the goal of ideal GC from the perspective of GNN optimization consistency, and then a generalized GC optimization objective is derived, by which those traditional GC methods can be viewed nicely as special cases of this optimization paradigm. Based on this, **Pre**-trained **G**raph **C**ondensation (**PreGC**) via optimal transport is proposed to transcend the limitations of task- and architecture-dependent GC methods. Specifically, a hybrid-interval graph diffusion augmentation is presented to suppress the weak generalization ability of the condensed graph on particular architectures by enhancing the uncertainty of node states. Meanwhile, the matching between optimal graph transport plan and representation transport plan is tactfully established to maintain semantic consistencies across source graph and condensed graph spaces, thereby freeing graph condensation from task dependencies. To further facilitate the adaptation of condensed graphs to various downstream tasks, a traceable semantic harmonizer from source nodes to condensed nodes is proposed to bridge semantic associations through the optimized representation transport plan in pre-training. Extensive experiments verify the superiority and versatility of PreGC, demonstrating its task-independent nature and seamless compatibility with arbitrary GNNs.

## 1   Introduction

Graph neural networks (GNNs) have emerged as a powerful approach to graph data analysis, showing excellent efficacy in a variety of domains, such as recommender systems [5], text analysis [56, 53], social network analysis [52], and so on. However, the dual challenges of exponentially growing graph data volumes [58] and increasingly sophisticated GNN architectures [63] present significant obstacles for GNN training. These challenges become particularly acute in scenarios that require multiple GNN training, such as continual graph learning [21] and federated graph learning [30].

In response to the above issues, graph condensation (GC) [25, 64] is proposed to optimize and synthesize a small-scale condensed graph from the real graph. The core objective is to ensure that

---

[*]Corresponding author

condensed graphs achieve comparable performance as original graphs when training the same GNN. This approach significantly reduces computational overhead and becomes an up-and-coming solution for efficient graph learning. Depending on matching strategies, GC fall mainly into three types. For example, [25, 24, 55] distilled the properties of the real graph by mimicking its gradient changes in GNN training. [64, 59] replicated the training trajectory on original graphs to synthesize condensed graphs. Other studies [32, 16] generated the graph by aligning the latent space representations of both graphs and maintaining the distribution consistency. Recently, [16] pointed out that all these methods can be generalized as distribution matching, with the basic pipeline shown in Fig. 1 (a).

Despite these significant efforts, current methods suffer from two critical limitations: **(i) Architecture-bound optimization.** Current methods can be regarded as condensing through constraints in the same representation space [16]. Therefore, they have to adopt a fixed encoder to map nodes from the graph space to the representation space in condensation. Such an excessive dependence on a specific architecture can degrade the generalization of condensed graphs, leading to poor performance on other models. In addition, the learnable model parameters inherit task bias [46], which causes another challenge. **(ii) Task-constrained condensation.** To simplify optimization [25], existing methods typically assume that tasks and labels are known, thus the effectiveness of GC depends heavily on

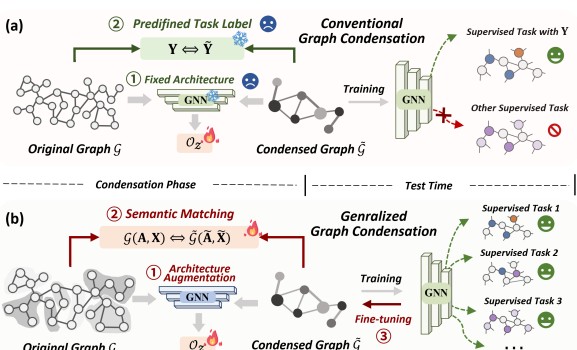

Figure 1: (a) The conventional GC framework. Due to specific ① architecture and ② task dependencies, $\tilde{\mathcal{G}}$ lacks generalization and reusability. (b) Our proposed generalized GC framework. It solves the above problems by ① and ②, and fine-tuning $\tilde{\mathcal{G}}$ in specific scenarios by ③.

the guidance of supervision signals. Unfortunately, this assumption often proves impractical in real-world scenarios. For example, in social networks, multiple tasks may coexist (e.g., preference classification, income prediction, and relation prediction) [12], and user labels frequently vary owing to privacy constraints and annotation discrepancies [30]. When the task or label changes, conventional GC methods require re-condensation to capture new knowledge, significantly diminishing the reusability of condensed graphs. Although [15] attempted to address label dependency by contrastive learning, its reliance on class similarity hinders its application to other supervised tasks.

To tackle the two critical challenges altogether, we first revisit the GC objective based on the goal of condensed graphs. It aims to achieve comparable performance to the original graph when training the same GNN (i.e. GNN optimization consistency). From this perspective, we derive a generalized GC optimization paradigm without task constraints as shown in Fig. 1 (b) and design a flexible and generalized GC method termed **Pre**-trained **G**raph **C**ondensation (PreGC). Inspired by data augmentation [10], graph diffusion augmentation is proposed to perturb node diffusion states via hybrid intervals, thereby enhancing the generalizability of condensed graphs and mitigating their dependence on specific architectures. To decouple the reliance on task labels, we present a novel matching mechanism, named transport plan matching. It maintains consistency in semantic associations between graph space and representation space, thus guaranteeing a unique mapping from each source node to its condensed counterpart. More importantly, transport plans provide explicit feedback on the significance of the source node, alleviating the poor traceability and interpretability issues in conventional condensation methods. After pre-training, a traceable semantic harmonizer via the optimal transport plan is derived, enabling flexible transfer of task signals from the original graph to the condensed graph for adaptation to diverse tasks.

**To the best of our knowledge**, PreGC is the first work for generalized graph condensation considering both architecture- and task-agnostic scenarios. Our main contributions are outlined as follows:

- Formal theoretical analysis reveals that conventional GC methods are inherently governed by the reconstruction term and the fitting term. Inspired by such an insight, we propose a pre-training framework of graph condensation, generalizing the condensed graph to more scenarios.

- To empower the condensed graph for architecture-agnostic scenarios, a hybrid-interval graph diffusion augmentation is presented, which enhances the diversity of node representations by introducing stochasticity into the diffusion process of both the condensed and original graphs.

- We propose an optimal transport plan matching to realize semantic alignment without task constrained. By keeping a consistent optimal plan for graph alignment and representation alignment, the uniqueness of semantic associations between source nodes and condensed nodes is guaranteed.

- Experiments on four mainstream tasks and nine representative GNN architectures demonstrate the superiority and generalization of PreGC. Further data valuation of the original graph highlights the PreGC's excellent traceability and interpretability.

## 2 Preliminary

**Graph Condensation (GC).** Given a large-scale original graph $\mathcal{G} = (\mathbf{A}, \mathbf{X})$ with a node set $\mathcal{V}$ ($|\mathcal{V}| = N$), where $\mathbf{A} \in \mathbb{R}^{N \times N}$ denotes the symmetrically normalized adjacency matrix and $\mathbf{X} \in \mathbb{R}^{N \times f}$ is a $f$-dimensional node feature matrix. GC [25, 33] aims to condense a small synthetic graph $\tilde{\mathcal{G}} = (\tilde{\mathbf{A}}, \tilde{\mathbf{X}})$ with a condensed node set $\tilde{\mathcal{V}}$ ($|\tilde{\mathcal{V}}| = M$) form the real large graph $\mathcal{G}$, where $\tilde{\mathbf{A}} \in \mathbb{R}^{M \times M}$, $\tilde{\mathbf{X}} \in \mathbb{R}^{M \times f}$, and $M \ll N$. At the same time, trained GNNs with the specific task have comparable performance on both $\tilde{\mathcal{G}}$ and $\mathcal{G}$, thus accelerating the training of GNNs.

**Graph Optimal Transport.** Optimal transport (OT) [39] aims to seek the most cost-efficient transport plan $\pi$ that transforms a source distribution $\mu$ into a target distribution $\nu$ while minimizing the total transport cost (i.e., the optimal transport distance). The elements of $\pi$ represent the probability of mass transferring from one location to another. In this work, we primarily concentrate on graph optimal transport [6, 43], which extends the OT to compare structured data. Given two graphs $\mathcal{G} = (\mathbf{A}, \mathbf{X})$ and $\tilde{\mathcal{G}} = (\tilde{\mathbf{A}}, \tilde{\mathbf{X}})$ with the nodes' empirical distribution $\mu \in \mathbb{R}^N$ and $\nu \in \mathbb{R}^M$, the general form of the graph optimal transport distance can be formalized as:

$$\mathcal{W}(\mathcal{G}, \tilde{\mathcal{G}}) = \min_{\pi \in \Pi(\mu, \nu)} \left\langle \mathcal{C}(\mathbf{X}, \tilde{\mathbf{X}}, \mathbf{A}, \tilde{\mathbf{A}}, \pi), \pi \right\rangle \tag{1}$$

where $\Pi(\mu, \nu) = \{\pi \in \mathbb{R}^{N \times M} | \pi \mathbf{1}_M = \mu, \mathbf{1}_N \pi = \nu\}$ is set of the joint distributions $\pi$ with marginals $\mu$ and $\nu$, and and $\langle \cdot, \cdot \rangle$ denotes the inner product for matrices. $\mathcal{C}$ represents the cost function that quantifies the cost of transporting mass from elements in one domain to another.

The transport distance quantifies the distribution divergence between condensed and original graphs, offering a novel optimization approach for GC. Meanwhile, the transport plan establishes explicit node correspondences, overcoming the limitations of poor traceability in conventional GC methods.

## 3 Revisiting and Generalizing Graph Condensation

In this section, we revisit the objective of graph condensation and reveal the limitations of existing GC methods, thus motivating the design of PreGC.

**Revisiting.** For convenience, we follow [33] and start with a vanilla example, which adopts a $K$-order SGC [48] as the GNN and simplifies the objective of GNNs into the MSE loss: $\mathcal{L}_{MSE} = \|\mathbf{A}^K \mathbf{X} \mathbf{W} - \mathbf{Y}\|_F^2$, where $\mathbf{Y} \in \mathbb{R}^{N \times C}$ denotes the target variable and $\mathbf{W} \in \mathbb{R}^{f \times C}$ is the model parameter. For the original graph $\mathcal{G}$, the optimal $\mathbf{W} = \arg\min_{\mathbf{W}} \|\mathbf{A}^K \mathbf{X} \mathbf{W} - \mathbf{Y}\|_F^2$ can be obtained by MSE loss. In general, the goal of condensation is that the GNN trained on the condensed graph $\tilde{\mathcal{G}}$ to achieve comparable performance to those trained on $\mathcal{G}$, i.e

$$\min \|\mathbf{W} - \tilde{\mathbf{W}}\|_F = \min \|(\mathbf{A}^K \mathbf{X})^\dagger \mathbf{Y} - (\tilde{\mathbf{A}}^K \tilde{\mathbf{X}})^\dagger \tilde{\mathbf{Y}}\|_F \tag{2}$$

where $\tilde{\mathbf{W}} = \arg\min_{\tilde{\mathbf{W}}} \|\tilde{\mathbf{A}}^K \tilde{\mathbf{X}} \tilde{\mathbf{W}} - \tilde{\mathbf{Y}}\|_F^2$ with target variable $\tilde{\mathbf{Y}} \in \mathbb{R}^{M \times C}$ on $\tilde{\mathcal{G}}$, and $\dagger$ denotes pseudo inverse. Subsequently, we further derive the following proposition (Proofs are in Appendix A):

**Proposition 3.1.** *Suppose that it has an analytical filter $g(\cdot)$ for a GNN model, the performance approximation error $\mathcal{O}_{\mathcal{W}}$ of condensed graph is jointly bounded by the reconstruction term $\mathcal{O}_{\mathcal{Z}}$ and the fitting term $\mathcal{O}_{\mathcal{Y}}$.*

$$\underbrace{\|\mathbf{W} - \tilde{\mathbf{W}}\|_F}_{\mathcal{O}_{\mathcal{W}}} \leq \|\boldsymbol{\mathcal{M}}_{\mathcal{Z}}(\mathbf{Y})\|_F \cdot \underbrace{\|\boldsymbol{\mathcal{M}}_{\mathcal{Z}}(g(\mathbf{L})\mathbf{X})^\dagger - (g(\tilde{\mathbf{L}})\tilde{\mathbf{X}})^\dagger\|_F}_{\mathcal{O}_{\mathcal{Z}}} + \underbrace{\|\boldsymbol{\mathcal{M}}_{\mathcal{Z}}(\mathbf{Y}) - \tilde{\mathbf{Y}}\|_F}_{\mathcal{O}_{\mathcal{Y}}} \cdot \|(g(\tilde{\mathbf{L}})\tilde{\mathbf{X}})^\dagger\|_F \tag{3}$$

where $\mathcal{M}_{\mathcal{Z}}(\cdot)$ is any mapping function that aligns $g(\tilde{\mathbf{L}})\tilde{\mathbf{X}}$ and $g(\mathbf{L})\mathbf{X}$ or $\tilde{\mathbf{Y}}$ and $\mathbf{Y}$. $\mathbf{L} = \mathbf{I}_N - \mathbf{A}$ and $\tilde{\mathbf{L}} = \mathbf{I}_M - \tilde{\mathbf{A}}$ are the Laplacian matrices of $\mathcal{G}$ and $\tilde{\mathcal{G}}$.

Proposition 3.1 reveals that the reconstruction term $\mathcal{O}_{\mathcal{Z}}$ and the fitting term $\mathcal{O}_{\mathcal{Y}}$ are the two key factors for ideal GC. Unfortunately, current GC methods rely on specific GNNs (i.e. the rigid filter $g(\cdot)$), resulting in spectrally limited condensed graphs with higher reconstruction errors for $\mathcal{O}_{\mathcal{Z}}$. Moreover, the pre-defined task signal in condensation restricts $\mathcal{O}_{\mathcal{Y}}$ to the specific task. Consequently, the condensed graph can only achieve satisfactory performance with the pre-defined GNNs and task, severely limiting its generalizability and reusability. To overcome these limitations, we focus on proposition 3.1 and relax the above conditionally constrained GC objective into a generalized one:

**Definition 3.1.** *Generalized Graph Condensation. Given an original graph $\mathcal{G}$, the objective of ideal graph condensation is to seek a condensed graph $\tilde{\mathcal{G}}$ that simultaneously minimizes the representation-level and the semantic-level discrepancies relative to $\mathcal{G}$:*

$$\tilde{\mathcal{G}}^* = \arg\min_{\tilde{\mathcal{G}}} \Big\{ \underbrace{\Delta_{\mathcal{Z}}(\mathbf{Z}, \tilde{\mathbf{Z}})}_{\mathcal{O}_{\mathcal{Z}}} + \xi \underbrace{\Delta_{\mathcal{Y}}(\mathcal{G}, \tilde{\mathcal{G}})}_{\mathcal{O}_{\mathcal{Y}}} \Big\} \tag{4}$$

*where $\Delta_{\mathcal{Z}}(\cdot, \cdot)$ measures the representation-level discrepancy between representations $\mathbf{Z}$ and $\tilde{\mathbf{Z}}$ of $\mathcal{G}$ and $\tilde{\mathcal{G}}$. $\Delta_{\mathcal{Y}}(\cdot, \cdot)$ is a distance function measuring semantic-level discrepancy, and $\xi$ serves as a trade-off parameter balancing the fitting and reconstruction terms.*

Definition 3.1 presents a general GC optimization paradigm, of which current GC methods can be regarded as variant forms (Appendix C.2 shows more details). In this formulation, the generalized node representation $\mathbf{Z}$ replaces $g(\mathbf{L})\mathbf{X}$, and the task-specific fitting term $\mathcal{O}_{\mathcal{Y}}$ is relaxed to semantic alignment between two graphs. Therefore, by boosting the diversity of the representations $\mathbf{Z}$ and $\tilde{\mathbf{Z}}$, and designing unsupervised signals for $\Delta\mathcal{Y}$, it is possible to readily establish a pre-trained graph condensation framework. Such a framework enhances the generalization of the condensed graph and augments its adaptability across diverse scenarios.

## 4 Methodology

Fig. 2 illustrates the framework of PreGC, which strictly follows the GC objective in Eq. (4). In pre-training phase, graph diffusion augmentation is proposed to increase the diversity of representations in $\mathcal{O}_{\mathcal{Z}}$ to alleviate architectural dependencies. Subsequently, to realize task-agnostic graph condensation, PreGC adopts an innovative transport plan matching to align semantic associations in $\mathcal{O}_{\mathcal{Y}}$. In addition, test-time fine-tuning further achieves dynamic adaptation on specific tasks and architectures.

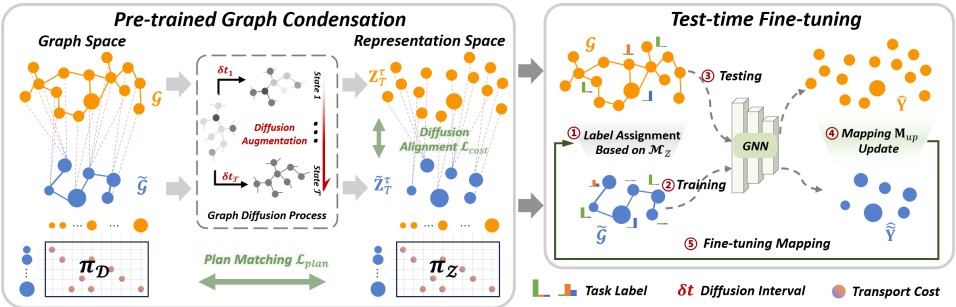

Figure 2: Overall pipeline of the proposed PreGC framework. $\pi_{\mathcal{D}}$ and $\pi_{\mathcal{Z}}$ denote optimal transport plans for graph alignment and for representation alignment, respectively.

### 4.1 Diffusion Augmentation via Graph Heat Equation

As analyzed in Section 3, most GC methods adopt $K$-order SGC [48] or GCN [26] for encoding graphs to obtain $\mathbf{Z}$ and $\tilde{\mathbf{Z}}$ [42]. However, parameters in neural networks may inherit task-induced biases, thus distorting the intrinsic representation properties. Although GDEM [33] and CGC [16] attempt to use parameter-free encoders to obtain node representations, they remain architecture-bound, resulting in condensed graphs hard to generalize to other GNNs. More critically, under discrete

propagation, the representation distributions of two graphs evolve at markedly divergent rates due to diameter disparities, inevitably exacerbating the difficulty of GC.

The graph heat equation (GHE) [9], as a generalization of the diffusion equation on graph data (graph diffusion), serves as an effective solution to the above problems due to its non-parametric and dynamic continuity properties. Formally, GHE can be defined as $\frac{d\mathbf{Z}_t}{dt} = -\mathbf{L}\mathbf{Z}_t$, where $\mathbf{Z}_t$ is the evolved input representation at time $t$, and $\mathbf{Z}_0 = \mathbf{X}$. In our proposed PreGC, graph diffusion [31] is used to encode node representations and solved using Explicit Euler method:

$$\mathbf{Z}_{t+\delta t} = \mathbf{Z}_t - \delta t \mathbf{L}\mathbf{Z}_t = \mathbf{A}^{(\delta t)}\mathbf{Z}_t \tag{5}$$

where $\delta t$ is diffusion interval and $\mathbf{A}^{(\delta t)} = (1 - \delta t)\mathbf{I} + \delta t\mathbf{A}$. Finally, the representation (or state) $\mathbf{Z}_T$ at the terminal time $T = K \cdot \delta t$ can be formulated as $\mathbf{Z}_T = [\mathbf{A}^{(\delta t)}]^K \mathbf{X}$. Adjusting $\delta t$ enables smoother propagation rate modulation, mitigating representation instability in discrete propagation.

**Graph Diffusion Augmentation.** Indeed, deterministic diffusion states capture specific graph spectral responses. To eliminate architectural bias, the condensed graph should maintain state consistency with the original graph for arbitrary diffusion times to inherit the complete spectral properties.

As a sample generation strategy, data augmentation [10, 29] aims to enhance the generalizability and robustness of the model through diverse samples. Inspired by this, a simple yet efficient diffusion augmentation is proposed to improve the generalizability of the condensed graph by generating diverse diffusion states with hybrid intervals. This process can be formulated as:

$$\mathcal{S}(\mathbf{Z}_T^\tau) = \left\{ \mathbf{Z}_T^\tau \mid \mathbf{Z}_T^\tau = [\mathbf{A}^{(\delta t_\tau)}]^K \mathbf{X}, \ \delta t_\tau \sim \mathcal{P}, \ \forall \delta t_\tau \leq \frac{2}{\lambda_{\max}(\mathbf{L})} \right\}_{\tau=0}^{\mathcal{T}} \tag{6}$$

where $\mathcal{P} = \mathcal{U}(\delta t_{\min}, 2/\lambda_{\max}(\mathbf{L}))$, $\mathcal{T}$ denotes the condensation epoch, and $\lambda_{\max}(\mathbf{L})$ is the maximum eigenvalue of $\mathbf{L}$. $\forall \delta t_\tau \leq 2/\lambda_{\max}(\mathbf{L})$ ensures the stability of the graph diffusion. Similarly, $\mathcal{S}(\tilde{\mathbf{Z}}_T^\tau)$ is obtained by keeping the same $\delta t_\tau$ as above on $\tilde{\mathcal{G}}$.

It can be seen from Eq. (6) that the stochastic variation of the diffusion interval $\delta t$ raises uncertainty of the diffusion state at time $T$. By aligning the two graph representations $\mathbf{Z}_T^\tau$ and $\tilde{\mathbf{Z}}_T^\tau$, the condensed graph is guided to capture the diffusion trajectories of the original graph across varying states. In addition, we can derive the following proposition (Proofs are in Appendix A):

**Proposition 4.1.** *When enough diffusion times $\{\delta t_\tau\}_{\tau=1}^{\mathcal{T}}$ are sampled from the distribution $\mathcal{P}$ over the interval $I = [\delta t_{\min}, 2/\lambda_{\max}(\mathbf{L})]$, the spectral response function is satisfied by sampling coverage over the entire spectral range:*

$$\lim_{\mathcal{T} \to \infty} \Pr(\max_i \sup_{\delta t \in I} \min_\tau |\Phi_i(\delta t) - \Phi_i(\delta t_\tau)| < \varphi) = 1, \ \forall \varphi > 0 \tag{7}$$

*where $\Phi_i(\delta t) = e^{-K\delta t \lambda_i}$ is the spectral response function for the $i$-th eigenvalue $\lambda_i$.*

Proposition 4.1 establishes a fundamental probabilistic guarantee for the completeness of spectral coverage in graph diffusion augmentation. This strategy breaks through the fixed-architecture constraints in conventional GC methods and effectively preserves the original spectral properties.

## 4.2 Optimal Transport Plan Matching

For the generalized GC objective in Eq. (4), the distance function $\Delta_{\mathcal{Y}}(\cdot, \cdot)$ is challenging to define without any task signal. However, semantic information should be an intrinsic property of the graph, rather than being influenced by the downstream task. In other words, the semantics of the same node should remain unchanged in both the graph and the representation spaces. This insight inspired us to ensure the consistency of semantic associations between $\mathcal{G}$ and $\tilde{\mathcal{G}}$ by defining two similarity metrics in graph and representation spaces, respectively. In fact, it is nontrivial to directly compare these two metrics, since they belong to two different spaces and are two disjoint objects by nature.

Fortunately, optimal transport theory provides an elegant framework for this purpose. Therefore, we present an optimal transport plan matching, which aims to find the consistent optimal plan for graph alignment and representation alignment. For the graph space, a natural idea is to consider both node features and structure information in the transport plan. In PreGC, the fused Gromov-Wasserstein [43] is adopted to obtain the optimal transport plan $\pi_{\mathcal{D}}^* \in \mathbb{R}^{N \times M}$:

$$\pi_{\mathcal{D}}^*(\mathcal{G}, \tilde{\mathcal{G}}) = \operatorname*{arg\,min}_{\pi_{\mathcal{D}} \in \Pi(\mu, \nu)} \left\langle \gamma \mathcal{K}(\mathbf{X}, \tilde{\mathbf{X}}) + (1 - \gamma)\mathcal{J}(\mathbf{A}, \tilde{\mathbf{A}}) \otimes \pi_{\mathcal{D}}, \pi_{\mathcal{D}} \right\rangle \tag{8}$$

where $\otimes$ denotes the tensor-matrix multiplication, and $\gamma$ represents the coefficient used to control the importance of structure and features. $\mathcal{K}(\mathbf{X}, \tilde{\mathbf{X}})_{i,j} = \|\mathbf{X}_i - \tilde{\mathbf{X}}_j\|_2$ and $\mathcal{J}(\mathbf{A}, \tilde{\mathbf{A}})_{ij,kl} = |\mathbf{A}_{ik} - \tilde{\mathbf{A}}_{jl}|$ are cost metric functions that measure feature discrepancy and structure discrepancy, respectively. For the node presentation, we leverage Wasserstein distance to obtain the optimal plan $\pi_{\mathcal{Z}}^* \in \mathbb{R}^{N \times M}$:

$$\pi_{\mathcal{Z}}^*(\mathbf{Z}_T^\tau, \tilde{\mathbf{Z}}_T^\tau) = \underset{\pi_{\mathcal{Z}} \in \Pi(\mu,\nu)}{\arg\min} \left\langle \mathcal{K}(\mathbf{Z}_T^\tau, \tilde{\mathbf{Z}}_T^\tau), \pi_{\mathcal{Z}} \right\rangle \tag{9}$$

Intuitively, the optimal transport plan characterizes the transferring probability from source nodes to condensed nodes, and semantic association consistency in different spaces is preserved by minimizing the divergence between two transport plans. The transport plan matching loss is defined as follows:

$$\mathcal{L}_{plan} = \Delta_{\mathcal{Y}}\left(\pi_{\mathcal{D}}^*(\mathcal{G}, \tilde{\mathcal{G}}), \pi_{\mathcal{Z}}^*(\mathbf{Z}_T^\tau, \tilde{\mathbf{Z}}_T^\tau)\right) \tag{10}$$

Without loss of generality, $\Delta_{\mathcal{Y}}(\cdot, \cdot)$ is set to the Frobenius norm in this paper.

### 4.3 Node Significance Evaluation

Unlike conventional GC works without explicittly considering node contributions for condensation, the optimal resresentation transport plan $\pi_{\mathcal{Z}}^*$ here not only delineates the semantic associations between both graphs, but also inversely reflects the potential significance of the source nodes. Specifically, let's first define a mask matrix $\pi_{mask} \in \mathbb{R}^{N \times M}$ to filter out the most influential nodes from source graph $\mathcal{G}$ on each condensed node $j$ , i.e., we have $(\pi_{mask})_{i,j} = (\pi_{\mathcal{Z}}^*)_{i,j}$ when $(\pi_{\mathcal{Z}}^*)_{i,j}$ is among the top-$H$ of the $j$-th column, otherwise $(\pi_{mask})_{i,j} = 0$. Then, the node significance score vector $\mathbf{s} \in \mathbb{R}^N$ of the source graph can be obtained by:

$$\mathbf{s} = \pi_{mask} \cdot \mathbf{1}_M \tag{11}$$

where $\mathbf{1}_M \in \mathbb{R}^M$ denotes the all-ones vector. Essentially, the significance $\mathbf{s}$ explicitly reveals the contributions of different source nodes for the condensed graph. Noticeably, the evaluation of node significance based on $\mathbf{s}$ are two-folds. On the one hand, it provides more transparent interpretability for graph condensation, and on the other hand, it also helps to guide node labeling effectively in an active learning way [20]. Section 5.2 further confirms the validity of the node significance evaluation.

### 4.4 Condensed Graph Optimization

**Pre-training.** Benefiting from Eq. (9), the reconstruction term $\mathcal{O}_{\mathcal{Z}}$ can be achieved by diffusion alignment between $\mathbf{Z}_T^\tau$ and $\tilde{\mathbf{Z}}_T^\tau$. Therefore, according to Eq. (4), the total loss can be denoted as:

$$\mathcal{L}_{total} = \mathcal{L}_{cost} + \xi \mathcal{L}_{plan}, \quad \mathcal{L}_{cost} = \Delta_{\mathcal{Z}}(\mathbf{Z}_T^\tau, \tilde{\mathbf{Z}}_T^\tau) \tag{12}$$

where $\Delta_{\mathcal{Z}}(\cdot, \cdot)$ denotes the Wasserstein distance [39]. After obtaining the condensed graph $\tilde{\mathcal{G}}$, the transport plan $\pi_{\mathcal{Z}}^*$, which reflects the transfer probabilities between the source nodes from $\mathcal{G}$ and the condensed nodes, is also retained. To ensure the uniqueness of the semantic mapping, we discretize $\pi_{\mathcal{Z}}^*$ to obtain the semantic assignment matrix $\mathbf{M} \in \mathbb{R}^{N \times M}$:

$$\mathbf{M}_{i,j} = \mathbb{I}(j = \underset{1 \le l \le M}{\arg\max} (\pi_{\mathcal{Z}}^*)_{i,l}) \tag{13}$$

When the condensed graph $\tilde{\mathcal{G}}$ needs to be trained on downstream tasks, the target variable $\tilde{\mathbf{Y}} \in \mathbb{R}^{M \times C}$ can be obtained from the original label by the traceable semantic harmonizer $\mathcal{M}_{\mathcal{Z}}(\cdot)$:

$$\tilde{\mathbf{Y}} = \mathcal{M}_{\mathcal{Z}}(\mathbf{Y}_{tr}) = \mathbf{D}_\Omega^{-1} \mathbf{M}_\Omega^\top \mathbf{Y}_{tr} \tag{14}$$

where $\mathbf{Y}_{tr} \in \mathbb{R}^{N_{tr} \times C}$ is a training label with $N_{tr}$ labelled nodes, and $\Omega = \{i_1, ..., i_{N_{tr}}\}$ is the set of their row indices. $\mathbf{M}_\Omega \in \mathbb{R}^{N_{tr} \times M}$ denotes the sub-assignment matrix defined by $(\mathbf{M}_\Omega)_{k,j} = \mathbf{M}_{i_k,j}$. $\mathbf{D}_\Omega = \mathrm{diag}(\mathbf{M}_\Omega^\top \mathbf{1}_{N_{tr}}) \in \mathbb{R}^{M \times M}$, and $\mathrm{diag}(\cdot)$ is used to create a diagonal matrix.

**Test Time Fine-tuning.** To further unlock the potential of condensed graphs while improving the performance of condensed graphs on different tasks and GNN architectures, we draw on the idea of fine-tuning pre-trained models [3] to propose PreGC$_{ft}$, a strategy for achieving finer semantic alignment by fine-tuning the assignment matrix. Specifically, given a decay rate $\epsilon \in [0, 1]$, the assignment matrix $\mathbf{M}$ is updated every $\tau_{up}$ epochs according to predefined tasks and GNN:

$$\mathbf{M} \leftarrow \epsilon \mathbf{M} + (1 - \epsilon) \mathbf{M}_{up} \tag{15}$$

where $\mathbf{M}_{up} \in \mathbb{R}^{N \times M}$ is obtained from $\tilde{\pi}_{\mathcal{Y}}^*(\hat{\mathbf{Y}}, \hat{\tilde{\mathbf{Y}}})$ similar as Eq. (9). $\hat{\mathbf{Y}}$ and $\hat{\tilde{\mathbf{Y}}}$ are the predicted outputs by the specific GNN encoded from $\mathcal{G}$ and $\tilde{\mathcal{G}}$, respectively.

Through the above optimization framework, we empower GC with pre-training and fine-tuning capabilities. The overall pipeline of PreGC are summarized in Appendix B.

## 5  Experiments

### 5.1  Experimental Settings

**Datasets.** To comprehensively evaluate the condensation performance of PreGC, five graph datasets (Cora, Citeseer, Pubmed [26], OGB-Arxiv [60], and H&M [2]) are utilized in experiments. Different tasks are tested to validate the generalization of PreGC, including node classification (NC), node clustering (NClu), link prediction (LP), and node regression (NR). Specifically, OGB-Arxiv includes two NC tasks: predicting paper topics (T) and publication years (Y). H&M includes an NC task: predicting product categories (C), and an NR task: predicting product prices (P). Two condensation ratios ($r = \frac{M}{N}$) are considered. More details are provided in Appendix C.1.

**Baselines & Implementations.** We compare the proposed PreGC with six representative graph condensation methods (GCDM [32], GCond [25], SFGC [64], SGDD [55], GDEM [33], and CGC [16]). Following [25, 42], the condensed graph is synthesized according to the default settings of the above methods, and a certain GNN (default is SGC [48]) is trained on this graph and evaluated on the test set of the original graph. Moreover, other eight representative GNNs (GCN [26], APPNP [27], $k$-GNN [38], GAT [44], GraphSAGE [19], SSGC [65], BernNet [23], and GPRGNN [8]) are selected to evaluate the effectiveness of PreGC. The details of baselines, task settings, condensation settings, and parameter settings are provided in Appendix C.2 and C.3.

Table 1: Performance comparison on different tasks (We report test accuracy (%) on NC, NMI (%) on NClu, and AUC (%) on LP tasks. **Bold entry** is the best result, underline marks the runner-up).

| Dataset | Task | Ratio | GCDM (2022) | GCond (2022) | SFGC (2023) | SGDD (2023) | GDEM (2024) | CGC (2025) | PreGC (Our) | Whole |
|---|---|---|---|---|---|---|---|---|---|---|
| **Cora** | NC | 1.3% | 61.5±1.2 | 79.0±1.3 | 75.4±0.5 | 79.2±0.6 | 80.8±0.4 | 79.0±0.8 | **81.1±0.3** | 80.8±0.8 |
| | | 2.6% | 69.2±0.4 | 79.0±0.6 | 79.2±1.4 | 79.2±1.4 | 80.7±0.1 | **81.8±0.6** | 81.6±0.9 | |
| | NC→NClu | 1.3% | 36.9±0.7 | 58.1±1.6 | 56.0±2.0 | **59.0±1.2** | 57.2±0.8 | 58.2±1.2 | 54.6±1.0 | 59.3±1.5 |
| | | 2.6% | 45.3±0.5 | 57.6±0.9 | 57.3±0.7 | 58.7±0.5 | 57.0±0.5 | 59.4±0.5 | **59.6±0.5** | |
| | LP | 1.3% | 56.8±0.7 | 56.6±3.0 | N/A | 57.0±3.5 | 66.9±0.6 | 65.5±0.8 | **75.3±2.2** | 81.9±4.8 |
| | | 2.6% | 57.2±1.8 | 62.2±1.6 | N/A | 60.4±2.2 | 59.7±2.6 | 46.5±4.1 | **79.2±2.2** | |
| | LP→NClu | 1.3% | 19.1±4.7 | 26.8±10.1 | N/A | 14.3±6.6 | 29.5±3.8 | 8.9±0.7 | **33.6±1.0** | 42.3±3.3 |
| | | 2.6% | 13.1±2.8 | 32.0±6.0 | N/A | 13.4±3.3 | **35.2±1.4** | 10.3±1.9 | 34.9±2.1 | |
| **Citeseer** | NC | 0.9% | 54.7±1.2 | 68.9±1.0 | 67.1±0.9 | 68.9±1.6 | **70.6±0.3** | 70.5±0.4 | **70.6±0.5** | 69.2±0.4 |
| | | 1.8% | 52.1±0.9 | 67.9±0.7 | 65.1±1.5 | 69.7±1.7 | 70.5±0.4 | 70.6±0.4 | **70.8±0.8** | |
| | NC→NClu | 0.9% | 25.3±0.7 | 41.0±2.7 | 42.4±1.8 | 43.0±1.3 | 42.8±0.7 | 40.6±1.7 | **43.8±1.4** | 42.4±0.5 |
| | | 1.8% | 20.8±2.5 | 38.1±0.8 | 42.9±0.9 | 43.5±0.7 | **45.2±0.4** | 40.7±1.8 | 43.9±0.6 | |
| | LP | 0.9% | 61.5±3.8 | 61.5±2.0 | N/A | 60.0±2.2 | 65.8±3.7 | N/A | **66.5±1.6** | 80.8±4.9 |
| | | 1.8% | 57.5±5.3 | 62.3±3.4 | N/A | 56.9±2.7 | 64.3±3.6 | 55.5±3.9 | **69.3±3.3** | |
| | LP→NClu | 0.9% | 7.2±0.9 | 20.3±3.3 | N/A | 10.1±1.2 | 21.0±2.5 | N/A | **32.7±0.7** | 33.4±3.8 |
| | | 1.8% | 9.3±3.0 | 14.7±3.0 | N/A | 9.2±1.4 | 16.9±3.3 | 7.2±1.4 | **33.1±1.2** | |
| **Pubmed** | NC | 0.08% | 71.4±1.6 | 72.9±0.6 | 77.4±0.2 | 78.4±0.4 | 77.8±0.1 | 78.7±0.3 | **79.9±0.4** | 78.8±0.2 |
| | | 0.15% | 60.8±0.8 | 71.6±1.4 | 78.1±0.4 | 78.6±0.5 | 76.8±0.6 | 78.4±0.5 | **81.0±0.5** | |
| | NC→NClu | 0.08% | 24.8±2.8 | 21.1±5.5 | 35.3±0.6 | 35.6±0.8 | 35.5±0.1 | **37.0±0.6** | 35.6±0.8 | 37.1±0.8 |
| | | 0.15% | 19.4±0.5 | 18.9±0.7 | 36.9±0.2 | 36.1±0.3 | 34.1±0.4 | 38.0±0.4 | **38.3±0.7** | |
| | LP | 0.08% | 72.1±2.9 | 72.9±3.8 | N/A | 72.9±3.1 | 43.5±1.6 | N/A | **82.9±2.0** | 95.0±1.0 |
| | | 0.15% | 68.9±4.2 | 73.7±3.5 | N/A | 68.4±3.2 | 56.1±2.3 | 84.8±2.0 | **86.3±2.4** | |
| | LP→NClu | 0.08% | 14.1±3.0 | 6.2±4.4 | N/A | 3.0±0.4 | 1.6±0.2 | N/A | **22.5±0.7** | 30.7±3.3 |
| | | 0.15% | 13.7±0.4 | 4.1±1.8 | N/A | 12.0±1.8 | 20.1±4.2 | 23.5±0.3 | **24.0±1.0** | |

### 5.2  Experimental Results

**Performance of PreGC on Various Tasks.** We first demonstrate the generalizability of the condensed graph obtained by PreGC as shown in Table 1. Clearly, PreGC achieves optimal performance in most cases. In particular, PreGC consistently outperforms baselines on LP, demonstrating that the condensed graph of PreGC can well capture the topological relationships in the original structure. SFGC is unable to perform LP task (i.e. "N/A") due to its emphasis on structure-free during

condensation, limiting its task reusability. In addition, Table 2 shows the generalization of PreGC under different supervision tasks and the flexibility in task migration. Owing to the limitations that need to initialize condensed labels by the original class labels, existing methods can't be directly applied to node regression (i.e. "N/A"). By virtue of its unsupervised matching strategy, PreGC becomes the first GC method capable of adapting to NR task. Regardless of the tasks, PreGC consistently achieved optimal performance. In addition, the fine-tuning strategy PreGC$_{ft}$ achieves 1.46% gain compared to PreGC on H&M dataset.

Table 2: Performance comparison to baselines on different supervised tasks ("Y→T" means condensation on the "Y" task and testing on the "T" task. We report test accuracy (%) on NC and R$^2$ (%) on NR tasks. **Imp.** indicates the performance improvement over the best baseline).

| | OGB-Arxiv ($r = 1.25\%$) | | | | H&M ($r = 2.5\%$) | | | |
|---|---|---|---|---|---|---|---|---|
| | T→T | Y→T | Y→Y | T→Y | C→C | P→C | P→P | C→P |
| Whole | 64.43 ± 0.24 | | 55.37 ± 0.21 | | 77.07 ± 0.26 | | 50.80 ± 0.64 | |
| GCDM | 33.08±1.04 | 46.48±2.40 | 44.20±0.84 | 46.17±1.25 | 57.15±0.90 | N/A | N/A | 26.08±0.76 |
| GCond | 56.74±0.65 | 41.38±1.65 | 49.25±0.67 | 42.52±2.12 | 64.57±0.53 | N/A | N/A | 15.29±1.14 |
| SFGC | 59.64±0.22 | 50.15±1.55 | 50.07±0.47 | 44.30±1.76 | 60.09±1.24 | N/A | N/A | 4.81±1.04 |
| SGDD | 58.68±0.82 | 27.80±2.09 | 41.70±2.25 | 42.30±1.58 | 61.85±0.39 | N/A | N/A | 18.85±2.13 |
| GDEM | 53.93±0.71 | 40.01±0.87 | 49.17±0.32 | 47.80±0.96 | 52.61±3.05 | N/A | N/A | 6.83±2.99 |
| CGC | 58.71±0.65 | 53.32±1.61 | 49.50±0.72 | 48.53±0.24 | 64.35±0.33 | N/A | N/A | 15.95±1.11 |
| **PreGC** | 60.55±0.43 | 59.62±0.47 | 52.34±0.30 | 51.49±0.36 | 69.06±0.25 | 68.86±0.29 | 34.39±0.93 | 33.17±1.05 |
| **Imp.** | 0.91↑ | 6.30↑ | 2.84↑ | 2.96↑ | 4.49↑ | - | - | 7.09↑ |
| **PreGC$_{ft}$** | 60.81±0.40 | 60.86±0.43 | 52.47±0.20 | 52.41±0.25 | 69.98±0.24 | 69.98±0.28 | 34.75±0.74 | 34.63±0.69 |
| **Imp.** | 1.17↑ | 7.54↑ | 2.97↑ | 3.88↑ | 5.41↑ | - | - | 8.55↑ |

**Generalizability of PreGC across Different GNNs.** Ideal condensed graphs should perform well in different GNNs, and to this end, each synthetic graph is evaluated by nine GNNs, and the results are shown in Table 3. It is evident that the PreGC exhibits the highest average accuracy, suggesting that the condensed graph extracted by PreGC can consistently benefit various GNNs. Furthermore, only SFGC and PreGC have average accuracy exceeding that of training on the raw graph. However, SFGC can lead to limited application scenarios due to its lack of explicit structure.

Table 3: Node classification performance across different GNNs on Pubmed ($r = 0.15\%$). Highlight marks the lossless results and **Avg.** is the average results of GNNs.

| | SGC | GCN | APPNP | $k$-GNN | GAT | SAGE | SSGC | Bern. | GPR. | **Avg.** |
|---|---|---|---|---|---|---|---|---|---|---|
| Whole | 78.9±0.2 | 78.2±0.4 | 79.2±0.4 | 77.0±0.8 | 77.2±0.4 | 77.0±0.5 | 79.0±0.3 | 77.4±0.5 | 79.0±0.3 | 78.1±1.0 |
| GCDM | 60.8±0.8 | 56.8±1.1 | 70.0±0.9 | 72.3±0.5 | 76.2±0.8 | 53.3±0.8 | 69.8±0.4 | 68.2±1.3 | 71.1±0.7 | 66.5±7.7 |
| GCond | 71.6±1.6 | 73.1±1.7 | 73.0±0.2 | 65.9±2.4 | 71.7±2.9 | 69.4±1.5 | 73.3±0.4 | 72.6±2.2 | 72.4±0.8 | 71.4±2.4 |
| SFGC | 78.1±0.4 | 78.9±0.7 | 79.7±0.6 | 78.1±0.7 | 77.5±0.8 | 78.1±0.3 | 78.7±0.3 | 78.4±0.4 | 79.0±0.2 | 78.3±1.1 |
| SGDD | 78.6±0.5 | 78.7±0.3 | 78.7±0.7 | 76.7±0.7 | 73.4±3.0 | 76.4±0.4 | 78.6±0.2 | 77.3±0.5 | 77.4±0.7 | 77.3±1.7 |
| GDEM | 76.8±0.6 | 77.3±0.3 | 77.7±0.4 | 78.2±0.5 | 76.7±0.7 | 76.4±1.3 | 77.5±1.1 | 76.9±0.4 | 77.0±0.3 | 77.2±0.6 |
| CGC | 78.4±0.5 | 79.5±0.3 | 78.8±0.4 | 75.7±0.9 | 78.9±0.5 | 76.4±0.4 | 79.0±0.4 | 77.3±0.4 | 78.5±0.6 | 78.1±1.3 |
| **PreGC** | 81.0±0.5 | 80.3±0.4 | 80.9±0.3 | 79.1±0.5 | 77.8±1.0 | 77.9±1.0 | 81.3±0.4 | 80.2±0.3 | 80.0±0.5 | 79.8±1.3 |

**Performance Gap between Condensed Graph and Original Graph.** According to [16, 49, 33], the quality of condensed graphs can be quantified through class-level features on NC task. Therefore, we present the labeled reconstruction error $LRE = \frac{1}{K} \sum_{k=0}^{K} \|\mathbf{Y}^\dagger \mathbf{A}^k \mathbf{X} - \tilde{\mathbf{Y}}^\dagger \tilde{\mathbf{A}}^k \tilde{\mathbf{X}}\|_F$ with the class label to evaluate condensed graphs. This formulation systematically captures multiscale receptive fields through the expectation over propagation steps to evaluate the distribution discrepancy. We set $K = 5$ and the case $k = 0$ is also taken into account, which reflects the difference between the condensed and original features. As shown in Fig. 3, compared with two representative GC methods [16, 33], the $LRE$ of PreGC is consistently lower than that of the other two methods on different data and tasks. Since PreGC does not use any task labels in condensation, this is more challenging than baselines and also demonstrates the effectiveness of traceable semantic harmonizer.

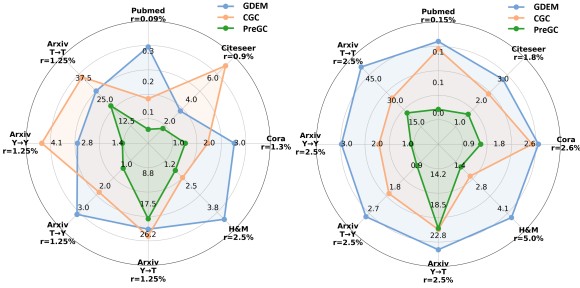

Figure 3: The $LRE$ of different GC methods.

**Data Valuation based on Node Significance.** To confirm the validity of the significance evaluation in Section 4.3, we rescreened the same number of nodes as the original training set based on Eq. (11) and retrained them on the original graph. Fig. 4 (b) shows a visualization of the distribution of nodes in the rescreened training set (Dark nodes denote the training set, and distinct colors denote different classes). Compared to the default training set provided by [26] (Fig. 4 (a)), the rescreened training set demonstrates greater coverage and distributional dispersion. We further measure distribution differences with the average nearest neighbor distance $\bar{d} = \frac{1}{|\mathcal{V}_{tr}|} \sum_{i \in \mathcal{V}_{tr}} \min_{j \neq i} \|\mathbf{z}_i - \mathbf{z}_j\|_2$ (larger $\bar{d}$ indicates wider coverage of the selected labels, and 2-order diffusion is considered, i.e., $\mathbf{Z} = \mathbf{A}^2\mathbf{X}$ by default), and the results also confirmed the above viewpoint. Fig. 4 (c) compares the performance of SGC trained with different training sets. Leveraging the rescreened nodes as supervision yields higher classification accuracy and markedly improves the training efficiency of the source graph.

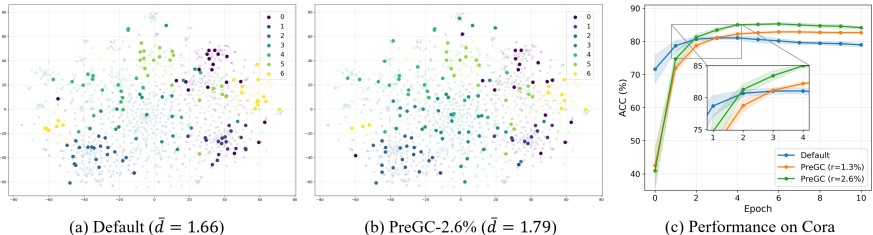

| (a) Default ($\bar{d} = 1.66$) | (b) PreGC-2.6% ($\bar{d} = 1.79$) | (c) Performance on Cora |

Figure 4: Performance of the training set selected via node significance on Cora.

Table 4: The prerformance of GC mehtods with different trainig ratio on OGB-Arxiv.

| Training Ratio | 0.15 | 0.3 | 0.45 | 0.6 | 0.75 | Count |
|---|---|---|---|---|---|---|
| Whole | 66.71±0.43 | 69.53±0.11 | 70.44±0.51 | 71.69±0.20 | 71.72±0.57 | - |
| GCDM | 35.98±1.31 | 36.32±1.08 | 36.09±0.89 | 36.94±0.60 | 38.35±0.67 | ×5 |
| GCond | 56.17±0.30 | 56.95±0.31 | 56.83±0.49 | 57.47±0.36 | 58.00±0.34 | ×5 |
| SFGC | 59.09±0.42 | 59.85±0.38 | 60.04±0.53 | 60.46±0.51 | 60.59±0.48 | ×5 |
| SGDD | 58.01±0.77 | 58.79±0.40 | 59.03±0.44 | 59.54±0.53 | 60.24±0.46 | ×5 |
| GDEM | 52.23±0.32 | 54.20±0.57 | 54.81±0.44 | 54.98±0.48 | 55.41±0.31 | ×5 |
| CGC | 58.39±0.33 | 59.04±0.49 | 58.92±0.72 | 59.53±0.56 | 60.05±0.29 | ×5 |
| **PreGC** | **60.53±0.32** | **61.96±0.56** | **61.72±0.63** | **63.37±0.49** | **63.58±0.82** | **×1** |

**Flexibility of PreGC.** The existing condensation methods are limited to concentrating under certain proportions of labels. When new labels are added in the original graph, these methods must re-condensation to capture this new knowledge in the condensed graph, which undoubtedly increases the condensation cost and limits its flexibility. PreGC can quickly transfer new labels to the condensed graph through the traceable semantic harmonizer without re-condensation. As shown in Table 4, we rescaled the training set ratio of the original graph and recorded the number of re-condensation (Count). With the structure and features of the condensed graph fixed, the condensed graph by PreGC shows similar patterns of change as the original graph as the training ratio increases. This label allocation strategy significantly improves the reusability of PreGC. Notably, existing baselines require repeated re-condensation to incorporate new label knowledge (×5), whereas PreGC achieves this with a single pass of semantic assignment matrix $\mathbf{M}$.

**Ablation Study.** To validate the efficacy of crucial optimization strategies, we conduct a comprehensive analysis on H&M dataset through ablation studies. The experimental results are shown in Table 5, wherein "PreGC$_{\text{w/o }Aug}$" is the PreGC that removes the graph diffusion augmentation and "PreGC$_{\text{w/o }\mathcal{L}_{cost}}$" is the adoption of

Table 5: Experimental results for the ablation study.

| H&M | $r = 2.5\%$ | | $r = 5.0\%$ | |
|---|---|---|---|---|
| | Category | Price | Category | Price |
| PreGC | 69.1±0.3 | 34.4±0.9 | 71.1±0.3 | 36.9±1.1 |
| PreGC$_{\text{w/o }Aug}$ | 66.6±0.7 | 32.2±0.9 | 67.9±0.4 | 33.3±1.4 |
| PreGC$_{\text{w/o }\mathcal{L}_{cost}}$ | 67.5±0.5 | 29.1±0.8 | 68.9±0.8 | 31.7±1.5 |

the existing strategy, i.e. predefined node labels before GC. We observe that both graph diffusion augmentation and $\mathcal{L}_{cost}$ contribute significantly to the improvement of the GC optimization. Notably, "PreGC$_{\text{w/o }\mathcal{L}_{cost}}$" induces label dependency in condensation, rendering the condensed graph via category prove inadequate for achieving superior performance on price prediction. Furthermore, the combination of these two modules increases the stability of the condensed graph.

More experimental analysis, including generalizability of PGC across different GNNs, comparison of $LRE$ with different GC methods, data valuation, and sensitivity analysis are detailed in Appendix D. In addition, we also discuss the related work and limitations in Appendix E.

# 6  Conclusions

In this work, we revisit the limitations of current GC methods and formulate a generalized GC objective. Following this paradigm, we further propose a pre-trained graph condensation framework for distilling a task-agnostic and architecture-flexible condensed graph. To avoid excessive dependence of condensed graphs on specific GNNs, a hybrid-interval graph diffusion augmentation is proposed to enhance generalization by increasing node state uncertainty. Subsequently, a plan matching mechanism is derived to eliminate task constraints, which relies on matching optimal transport plans for graph alignment and representation alignment to preserve semantic association consistency. In addition, supervised fine-tuning further refines the condensed task signals and improves the performance of the condensed graph on specific tasks. The experimental results consistently demonstrate the efficacy and reusability of PreGC, showcasing its generalization on various GNNs and tasks.

## Acknowledgments and Disclosure of Funding

This work was supported in part by the National Key Research and Development Program of China (2024YFF0505704), National Natural Science Foundation of China (62476022, 62506029), Beijing Natural Science Foundation (L251043, 4254085), and China Postdoctoral Science Foundation (2025M771589). The authors would also like to thank the anonymous reviewers for their constructive comments and valuable suggestions on this paper.

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

# Appendix

This is the appendix of our work: **"Towards Pre-trained Graph Condensation via Optimal Transport"**. In this appendix, we provide more details of the proposed PreGC in terms of theoretical proofs, dataset statistics, method analysis, experimental settings with some additional results, and related works.

## A    Proof of Propositions

**Proposition 3.1.** *Suppose that it has an analytical filter $g(\cdot)$ for a GNN model, the performance approximation error $\mathcal{O}_{\mathcal{W}}$ of condensed graph is jointly bounded by the reconstruction term $\mathcal{O}_{\mathcal{Z}}$ and the fitting term $\mathcal{O}_{\mathcal{Y}}$.*

$$\underbrace{\|\mathbf{W} - \tilde{\mathbf{W}}\|_F}_{\mathcal{O}_{\mathcal{W}}} \leq \|\mathcal{M}_{\mathcal{Z}}(\mathbf{Y})\|_F \cdot \underbrace{\|\mathcal{M}_{\mathcal{Z}}(g(\mathbf{L})\mathbf{X})^{\dagger} - (g(\tilde{\mathbf{L}})\tilde{\mathbf{X}})^{\dagger}\|_F}_{\mathcal{O}_{\mathcal{Z}}} + \underbrace{\|\mathcal{M}_{\mathcal{Z}}(\mathbf{Y}) - \tilde{\mathbf{Y}}\|_F}_{\mathcal{O}_{\mathcal{Y}}} \cdot \|(g(\tilde{\mathbf{L}})\tilde{\mathbf{X}})^{\dagger}\|_F$$

(16)

*where $\mathcal{M}_{\mathcal{Z}}(\cdot)$ is any mapping function that aligns $g(\tilde{\mathbf{L}})\tilde{\mathbf{X}}$ and $g(\mathbf{L})\mathbf{X}$ or $\tilde{\mathbf{Y}}$ and $\mathbf{Y}$. $\mathbf{L} = \mathbf{I}_N - \mathbf{A}$ and $\tilde{\mathbf{L}} = \mathbf{I}_M - \tilde{\mathbf{A}}$ are the Laplacian matrices of $\mathcal{G}$ and $\tilde{\mathcal{G}}$.*

*Proof.* Given a GNN with an analytical filtering function $g(\cdot)$, we have $g(\mathbf{L})\mathbf{X}\mathbf{W} = \mathbf{Y}$ on the original graph $\mathcal{G}$ and $g(\tilde{\mathbf{L}})\tilde{\mathbf{X}}\tilde{\mathbf{W}} = \tilde{\mathbf{Y}}$ on the condensed graph $\tilde{\mathcal{G}}$, respectively. Due to $\mathcal{M}_{\mathcal{Z}}(\cdot)$ can align the first dimensions, we also have $\mathcal{M}_{\mathcal{Z}}(g(\mathbf{L})\mathbf{X})\mathbf{W} = \mathcal{M}_{\mathcal{Z}}(\mathbf{Y})$ for $\mathcal{G}$. Therefore, the expression can be denoted as:

$$\|\mathbf{W} - \tilde{\mathbf{W}}\|_F$$
$$= \|\mathcal{M}_{\mathcal{Z}}(g(\mathbf{L})\mathbf{X})^{\dagger}\mathcal{M}_{\mathcal{Z}}(\mathbf{Y}) - (g(\tilde{\mathbf{L}})\tilde{\mathbf{X}})^{\dagger}\tilde{\mathbf{Y}}\|_F$$
$$= \|\mathcal{M}_{\mathcal{Z}}(g(\mathbf{L})\mathbf{X})^{\dagger}\mathcal{M}_{\mathcal{Z}}(\mathbf{Y}) - (g(\tilde{\mathbf{L}})\tilde{\mathbf{X}})^{\dagger}\mathcal{M}_{\mathcal{Z}}(\mathbf{Y}) + (g(\tilde{\mathbf{L}})\tilde{\mathbf{X}})^{\dagger}\mathcal{M}_{\mathcal{Z}}(\mathbf{Y}) - (g(\tilde{\mathbf{L}})\tilde{\mathbf{X}})^{\dagger}\tilde{\mathbf{Y}}\|_F \quad (17)$$
$$\leq \|(\mathcal{M}_{\mathcal{Z}}(g(\mathbf{L})\mathbf{X})^{\dagger} - (g(\tilde{\mathbf{L}})\tilde{\mathbf{X}})^{\dagger})\mathcal{M}_{\mathcal{Z}}(\mathbf{Y})\|_F + \|(g(\tilde{\mathbf{L}})\tilde{\mathbf{X}})^{\dagger}(\mathcal{M}_{\mathcal{Z}}(\mathbf{Y}) - \tilde{\mathbf{Y}})\|_F$$
$$\leq \|\mathcal{M}_{\mathcal{Z}}(\mathbf{Y})\|_F \cdot \|\mathcal{M}_{\mathcal{Z}}(g(\mathbf{L})\mathbf{X})^{\dagger} - (g(\tilde{\mathbf{L}})\tilde{\mathbf{X}})^{\dagger}\|_F + \|\mathcal{M}_{\mathcal{Z}}(\mathbf{Y}) - \tilde{\mathbf{Y}}\|_F \cdot \|(g(\tilde{\mathbf{L}})\tilde{\mathbf{X}})^{\dagger}\|_F$$

For the trajectory matching methods [64, 59], they integrated the above objectives into the process of graph condensation, that is:

$$\arg\min_{\tilde{\mathcal{G}}} \|\mathbf{W} - \tilde{\mathbf{W}}\|_F \Rightarrow$$
$$\arg\min_{\tilde{\mathcal{G}}} \Big\{ \|\mathcal{M}_{\mathcal{Z}}(\mathbf{Y})\|_F \cdot \underbrace{\|\mathcal{M}_{\mathcal{Z}}(g(\mathbf{L})\mathbf{X})^{\dagger} - (g(\tilde{\mathbf{L}})\tilde{\mathbf{X}})^{\dagger}\|_F}_{\mathcal{O}_{\mathcal{Z}}} + \underbrace{\|\mathcal{M}_{\mathcal{Z}}(\mathbf{Y}) - \tilde{\mathbf{Y}}\|_F}_{\mathcal{O}_{\mathcal{Y}}} \cdot \|(g(\tilde{\mathbf{L}})\tilde{\mathbf{X}})^{\dagger}\|_F \Big\}$$

(18)

According to [16], different matching strategies can be summarized into the same optimization objective. Thus, Proposion 3.1 shows that by minimizing the reconstruction term $\mathcal{O}_{\mathcal{Z}}$ and the fitting term $\mathcal{O}_{\mathcal{Y}}$, the performance derived from $\tilde{\mathcal{G}}$ can be approximated to that of $\mathcal{G}$.

This completes the proof of the proposion. $\qquad\square$

**Proposition 4.1.** *When enough diffusion times $\{\delta t_\tau\}_{\tau=1}^{\mathcal{T}}$ are sampled from the distribution $\mathcal{P}$ over the interval $I = [\delta t_{\min}, 2/\lambda_{\max}(\mathbf{L})]$, the spectral response function is satisfied by sampling coverage over the entire spectral range:*

$$\lim_{\mathcal{T} \to \infty} \Pr(\max_i \sup_{\delta t \in I} \min_\tau |\Phi_i(\delta t) - \Phi_i(\delta t_\tau)| < \varphi) = 1, \ \forall \varphi > 0 \tag{19}$$

*where $\Phi_i(\delta t) = e^{-K\delta t \lambda_i}$ is the spectral response function for the $i$-th eigenvalue $\lambda_i$.*

*Proof.* Given a graph $\mathcal{G} = (\mathbf{A}, \mathbf{X})$, the diffusion state with diffusion intervals $\delta t_\tau$ at time $T = K\delta t_\tau$ can be denoted as:

$$\mathbf{Z}_T^\tau = e^{-K\delta t_\tau \mathbf{L}}\mathbf{X} = \mathbf{U}e^{-K\delta t_\tau \mathbf{\Lambda}}\mathbf{U}^\top\mathbf{X} \tag{20}$$

For the $i$-th Laplacian eigenvalue $\lambda_i$ we write the corresponding spectral response $\Phi_i(\delta t) = e^{-K\lambda_i \delta t}$ with $i = 1, \dots, N$. Here, we begin with the following the lemma to better support Proposition 4.1:

**Lemma 1.** *For any two different diffusion intervals $\delta t_1, \delta t_2 \in I$, the difference in spectral response is satisfied:*

$$|\Phi_i(\delta t_1) - \Phi_i(\delta t_2)| = K\lambda_i e^{-K\psi_i\lambda_i} |\delta t_1 - \delta t_2| \tag{21}$$

*where $\psi_i$ lies between $\delta t_1$ and $\delta t_2$.*

*Proof.* According to the mean value theorem, for the function $\Phi_i(\delta t_\tau) = e^{-K\delta t_\tau \lambda_i}$, there exist $\psi_i \in [\min(\delta t_1, \delta t_2), \max(\delta t_1, \delta t_2)]$ such that:

$$\begin{aligned} \Phi_i(\delta t_2) - \Phi_i(\delta t_1) &= \Phi_i'(\psi_i)(\delta t_2 - \delta t_1) \\ \Phi_i'(\psi_i) &= -K\lambda_i e^{-K\psi_i\lambda_i} \end{aligned} \tag{22}$$

Taking absolute values on both sides of the equation:

$$|\Phi_i(\delta t_1) - \Phi_i(\delta t_2)| = |e^{-K\delta t_1\lambda_i} - e^{-K\delta t_2\lambda_i}| = K\lambda_i e^{-K\psi_i\lambda_i} \cdot |\delta t_1 - \delta t_2| \tag{23}$$

This completes the proof of the lemma. $\qquad\square$

From Lemma 1, each $\Phi_i$ is Lipschitz on $I$ with constant:

$$L_i = \sup_{\delta t \in I} |\Phi_i'(\delta t)| = K\lambda_i e^{-K\lambda_i \delta t_{\min}} \tag{24}$$

Let $L = \max_i L_i = K\lambda_N e^{-K\delta t_{\min}\lambda_N}$, where $\lambda_N = \lambda_{\max}(\mathbf{L})$. According to the lemma, for any $\delta t \in I$, if there exists a sampled point $\delta t_\tau$ such that $|\delta t - \delta t_\tau| < \frac{\varphi}{L}$, then for all $i$, the spectral response functions satisfy:

$$|\Phi_i(\delta t) - \Phi_i(\delta t_\tau)| = K\lambda_i e^{-K\psi_i\lambda_i} \cdot |\delta t - \delta t_\tau| \le L_i \cdot |\delta t - \delta t_\tau| \le L_i \cdot \frac{\varphi}{L} < \varphi \tag{25}$$

To ensure this condition, we divide the interval $I = [\delta t_{\min}, 2/\lambda_{\max}(\mathbf{L})]$ into $m = \lceil \frac{L \cdot |I|}{\varphi} \rceil = \lceil \frac{L(2/\lambda_{\max}(\mathbf{L}) - \delta t_{\min})}{\varphi} \rceil$ sub-intervals, each of length $\frac{|I|}{m} = \frac{2/\lambda_{\max}(\mathbf{L}) - \delta t_{\min}}{m} < \frac{\varphi}{L}$. Assuming the diffusion times $\{\delta t_\tau\}_{\tau=1}^{\mathcal{T}}$ are sampled from a distribution $\mathcal{P}$ with positive density over $I$, for each subinterval, the probability that at least one sampling point falls into it is:

$$p = 1 - (1 - p_{\min})^{\mathcal{T}} \tag{26}$$

where $p_{\min} > 0$ is the minimum probability of sampling a point in any subinterval. When $\mathcal{T} \to \infty$, we have $p \to 1$. The probability that all $m$ subintervals have at least one sampling point is $p^m$, and $p^m \to 1$ when $\mathcal{T} \to \infty$.

When all subintervals contain at least one sampling point, for any $\delta t \in I$, there exists a sampling point $\delta t_\tau$ within the same subinterval or in an adjacent subinterval, with $|\delta t - \delta t_\tau| < \frac{\varphi}{L}$. This ensures that:

$$\max_i \sup_{\delta t \in I} \min_\tau |\Phi_i(\delta t) - \Phi_i(\delta t_\tau)| < \varphi \tag{27}$$

Therefore:

$$\lim_{\mathcal{T} \to \infty} \Pr(\max_i \sup_{\delta t \in I} \min_\tau |\Phi_i(\delta t) - \Phi_i(\delta t_\tau)| < \varphi) = 1, \quad \forall \varphi > 0 \tag{28}$$

That is, when $\mathcal{T}$ is sufficiently large, the probability that the sampling points form a $\varphi$-net covering $I$ tends to 1, ensuring coverage of the entire function family $\{\Phi_i(\delta t) = e^{-K\delta t \lambda_i} : i = 1, 2, \dots, N\}$.

This completes the proof of the proposition. $\qquad\square$

# B Implementation Details of PreGC

## B.1 Algorithm

The detailed algorithm of PreGC is shown in Algorithm 1. In detail, the topology generator $\text{GEN}(\Theta)$ and the condensed features $\tilde{\mathbf{X}}$ are initialized before condensation. To accelerate the synthesis of the condensed graph, the clustering centers of K-means are adopted as the initialization of the condensed features. Here, $\text{GEN}(\Theta)$ can be any strategy used to generate graph structure, and

we adopt $\text{Sym}(\text{ReLU}(\frac{\Theta+\Theta^\top}{2}))$ with the symmetric normalization function $\text{Sym}(\cdot)$ for convenience. Then, node representations $\tilde{\mathbf{Z}}_T^\tau$ and $\mathbf{Z}_T^\tau$ are generated via the graph diffusion with stochastic states. By calculating the optimal transport of the two graphs in graph space and representation space, one can obtain the transport plans $\pi_{\mathcal{D}}^*$ and $\pi_{\mathcal{Z}}^*$ respectively, and the transport cost $\mathcal{L}_{cost}$. Plan matching loss can be obtained from the difference between metrics $\pi_{\mathcal{D}}^*$ and $\pi_{\mathcal{Z}}^*$. Finally, the PreGC minimizes the loss $\mathcal{L}_{total}$ by iteratively optimizing $\Theta$ and $\tilde{\mathbf{X}}$. Our work signifcantly differs from other GC methods, PreGC truly enables pre-training and provides transparent interpretability of the condensation.

---

**Algorithm 1** Pre-trained Graph Condensation (PreGC)

---

**Require:** Real graph $\mathcal{G} = (\mathbf{A}, \mathbf{X})$, learning rates $\eta_1$ and $\eta_2$
**Ensure:** A small-scale condensed graph $\tilde{\mathcal{G}} = (\tilde{\mathbf{A}}, \tilde{\mathbf{X}})$ with the mapping matrix $\mathbf{M}$.
 1: */* Condensation Phase */*
 2: Initialize $\Theta$ and $\tilde{\mathbf{X}}$.
 3: **for** $\tau = 0, 1, \cdots, \mathcal{T} - 1$ **do**
 4:     */* Graph Diffusion Augmentation */*
 5:     Compute $\tilde{\mathbf{A}} = \text{GEN}(\Theta)$; then $\tilde{\mathcal{G}} = (\tilde{\mathbf{A}}, \tilde{\mathbf{X}})$
 6:     Randomly generate diffusion interval $\delta t_\tau$
 7:     Graph diffusion on $\mathcal{G}$ and $\tilde{\mathcal{G}}$ to genenrate $\mathbf{Z}_T^\tau$ and $\tilde{\mathbf{Z}}_T^\tau$ according to Eq. (6), respectively
 8:     */* Transport Plan Matching */*
 9:     Calculate graph transport plan $\pi_{\mathcal{D}}^*$ according to Eq. (8)
10:     Calculate representaion transport plan $\pi_{\mathcal{Z}}^*$ and transport cost $\mathcal{L}_{cost}$ according to Eq. (9)
11:     Compute plan matching loss $\mathcal{L}_{plan}$ according to Eq. (10)
12:     */* Condensed Graph Updating */*
13:     Compute total loss $\mathcal{L}_{total} = \mathcal{L}_{cost} + \xi \mathcal{L}_{plan}$
14:     **if** $\tau \% (\tau_1 + \tau_2) < \tau_1$ **then**
15:         Update $\Theta \leftarrow \Theta - \eta_1 \nabla_\Theta \mathcal{L}_{total}$
16:     **else**
17:         Update $\tilde{\mathbf{X}} \leftarrow \tilde{\mathbf{X}} - \eta_2 \nabla_{\tilde{\mathbf{X}}} \mathcal{L}_{total}$
18:     **end if**
19: **end for**
20: Compute $\mathbf{M}$ according to Eq. (13) and $\tilde{\mathbf{A}} = \text{GEN}(\Theta)$
21: **return** $\tilde{\mathbf{A}}, \tilde{\mathbf{X}}$, and $\mathbf{M}$

---

## B.2 Details of Optimal Transport Solution

Since Eq. (9) can be regarded as a special form of Eq. (8) (i.e., $\gamma = 1$), our discussion primarily focuses on solving for $\pi_{\mathcal{D}}$ in Eq. (8).

Eq. (8) decomposes into two terms: the former is a Wasserstein distance that operates solely on node features. This term can be efficiently approximated using the Sinkhorn-Knopp algorithm [11], which iteratively converges to the optimal transport plan $\pi^*$. Specifically, the Sinkhorn-Knopp algorithm introduces an entropy regularizer and alternates between Sinkhorn projections, $\pi^{(0)} = \exp(-\mathcal{K}(\mathbf{X}, \tilde{\mathbf{X}})/\lambda)$ and $\pi^{(l+1)} = \mathcal{F}_1(\mathcal{F}_2(\pi^{(l)}))$, where $l$ represents the number of iterations and $\lambda$ denotes the weigth of regularization. The projections $\mathcal{F}_1(\pi) = \pi \oslash (\mathbf{1}\mathbf{1}^\top \pi)$ and $\mathcal{F}_2(\pi) = \pi \oslash (\pi \mathbf{1}\mathbf{1}^\top)$ denote the column normalization and row normalization, respectively, where $\oslash$ is element-wise division. They ensure that the rows and columns satisfy the given marginal distribution constraints. In the limit, this alternating projection will converge to a minimizer $\lim_{t \to \infty} \pi^{(t)} = \pi^*$.

The latter term corresponds to a Gromov-Wasserstein (GW) distance, which can be formulated as a nonconvex quadratic programming problem. We use semi-relaxed Gromov-Wasserstein divergence [45] and optimize $\pi$ with the conditional gradient solver. The gradient of $\pi$ with respect to the GW term is $\nabla_\pi = 2\mathcal{J}(\mathbf{A}, \tilde{\mathbf{A}}) \otimes \pi$. The conditional gradient algorithm consists in solving a linearization $\langle \mathbf{P}, \nabla_\pi \rangle$ at each iteration, where $\mathbf{P} \in \mathbb{R}^{N \times M}$ the extreme-point solution that arises from the linearized sub-problem solved. It can be solved by gradient descent with a direction $\mathbf{P}^{(l)} \pi^{(l)}$, followed by a line search for the optimal step.

In the optimization process, we utilize the third-party library functions GeomLoss and Python Optimal Transport (POT) to solve Eq. (8) and Eq. (9), respectively.

## C  Experimental Details

### C.1  Dataset Details

In this work, five graph datasets are adopted to evaluate the effectiveness of our condensation method. and their data statistics are shown in Table D1. Among them, three citation networks (Cora, Citeseer, and Pubmed) [26] are used for node classification (NC), graph clustering (GC), and link prediction (LP) tasks. OGB-Arxiv [60] is used to test the performance of GNNs on topic classification and publication year prediction tasks. H&M [2] is a subset that is obtained from a co-purchase network of products from the H&M company. The nodes are products whose features consist of the product's graphical appearance, perceived color, etc., and they are connected to edges if they are frequently purchased by the same user. H&M has two semi-supervised tasks: predict the categories of products (node classification) and predict their average prices (node regression).

Table D1: Details of dataset statistics. NC, NClu, LP, and NR denotes node classification, node clustering, link prediction, and node regression tasks respectively.

| Dataset | Nodes | Edges | Features | Task | Classes | Traing / Validation / Test |
|---|---|---|---|---|---|---|
| Cora | 2,708 | 5,429 | 1,433 | NC / NClu / LP | 7 | 140 / 500 / 1,000 |
| Citeseer | 3,327 | 4,732 | 3,703 | NC / NClu / LP | 6 | 120 / 500 / 1,000 |
| Pubmed | 19,717 | 44,338 | 500 | NC / NClu / LP | 3 | 60 / 500 / 1,000 |
| OGB-Arxiv | 14,167 | 33,520 | 128 | NC-Topic (T) | 40 | 4,585 / 2,517 / 7,065 |
|  |  |  |  | NC-Year (Y) | 5 | 7,125 / 3,492 / 3,550 |
| H&M | 15,508 | 920,284 | 191 | NC-Category (C) | 21 | 7,837 / 3,908 / 3,763 |
|  |  |  |  | NR-Price (P) | - | 7,724 / 3,843 / 3,941 |

### C.2  Baselines & GNNs

**Baselines**. In this work, six baselines have been selected for comparison.

- **GCDM** [32]: An efficient GC method that generates condensed graphs based on distribution matching by optimizing the maximum mean discrepancy between class representations.

- **GCond** [25]: A representation GC method that utilizes the gradient matching to align the model gradient from both graphs.

- **SFGC** [64]: The first structure-free GC method that synthesizes condensed features by trajectory matching while preserving the topological information of the original graph.

- **SGDD** [55]: A generalized graph condensation method with enhanced graph structure modeling by topological mapping.

- **GDEM** [33]: A GC method for topology enhancement by eigenbasis matching.

- **CGC** [16]: An efficient GC method with graph generation that introduces the training-free strategy in distribution matching.

To illustrate the distinctions among various approaches and emphasize our contributions, we present the relationship between six representative graph condensation methods and the generalized graph condensation paradigm, as shown in Table D2.

**GNN architectures**. To evaluate and compare the generalization of PreGC and other graph condensation methods, we test the node classification performance with nine different GNN architectures: SGC [48], GCN [26], APPNP [27], $k$-GNN [38], GAT [44], GraphSAGE (SAGE) [19], SSGC [65], BernNet (Bern.) [23], and GPRGNN (GPR.) [8]. Note that the focus of our work is on verifying the generalization of condensed graphs rather than the effectiveness of a certain GNN architecture. Therefore, we harmonize the GNN parameters for all datasets to ensure the fairness of the experiments. Concretely, the smoothing factor in APPNP, SSG, and GPR. is uniformly set to 0.1, the numer of attention heads in GAT is set to 4. For GNNs [27, 65, 23, 8] that can capture long-range dependencies, we set the number of propagation layers $K = 5$ by default and $K = 2$ otherwise. For other hyperparameters, we use the combination of parameters provided by PyG by default.

Table D2: Comparison of different graph condensation methods. For convenience, SGC $\tilde{\mathbf{A}}^K \tilde{\mathbf{X}}$ is uniformly set to GNN encoder in condensation, and $\tilde{\mathbf{A}}^{(\delta t)} = (1 - \delta t)\mathbf{I} + \delta t\tilde{\mathbf{A}}$.

| | Condensation Phase | | | | Testing Phase | |
|---|---|---|---|---|---|---|
| | Label Alignment | Graph Condensation | Optimization Objective | Representaion $\tilde{\mathbf{Z}}$ | Graph Interpretability | Task Adaptation |
| GCDM (Arxiv22) | Before | Distribution Matching | $\mathcal{O}_{\mathcal{Z}}$ | $\tilde{\mathbf{A}}\tilde{\mathbf{X}}\|\tilde{\mathbf{A}}^2\tilde{\mathbf{X}}\|\cdots\|\tilde{\mathbf{A}}^K\tilde{\mathbf{X}}$ | N/A | Classification |
| GCond (ICLR22) | Before | Gradient Matching | $\mathcal{O}_{\mathcal{W}}$ (w/o $\mathcal{O}_{\mathcal{Y}}$) | $\tilde{\mathbf{A}}^K\tilde{\mathbf{X}}$ | N/A | Classification |
| SFGC (NeurIPS23) | Before | Trajectory Matching | $\mathcal{O}_{\mathcal{W}}$ (w/o $\mathcal{O}_{\mathcal{Y}}$) | $\mathbf{I}^K\tilde{\mathbf{X}}$ | N/A | Classification |
| SGDD (NeurIPS23) | Before | Gradient Matching | $\mathcal{O}_{\mathcal{W}}$ (w/o $\mathcal{O}_{\mathcal{Y}}$) | $\tilde{\mathbf{A}}^K\tilde{\mathbf{X}}$ | Topology | Classification |
| GDEM (ICML24) | Before | Eigenbasis Matching | $\mathcal{O}_{\mathcal{Z}}$ | $\tilde{\mathbf{X}}^\top\tilde{\mathbf{L}}\tilde{\mathbf{X}}$ | Topology | Classification |
| CGC (WWW25) | Before | Distribution Matching | $\mathcal{O}_{\mathcal{Z}}$ | $\tilde{\mathbf{A}}^K\tilde{\mathbf{X}}$ | N/A | Classification |
| **PreGC** (Our) | After | Transport Plan Matching | $\mathcal{O}_{\mathcal{Z}}, \mathcal{O}_{\mathcal{Y}}$ | $[\tilde{\mathbf{A}}^{(\delta t)}]^K\tilde{\mathbf{X}}$ | Semantics | Any Tasks |

## C.3 Experimental Setup

**Task setting**. Unless otherwise noted, we use the default training/validation/testing ratios to evaluate the performance of the condensed graphs for supervised tasks. Specifically, for Whole , the training set of the original graph is used for training. For graph condensation methods, we train on the condensed graph, and validation and testing are always evaluated on the validation and test sets of the original graph. For link prediction task, we follow [42] and randomly select 10% ratio of edges as validation and test sets, respectively. In node clustering task, the K-means algorithm is adopted to the learned embeddings of all nodes (For example, "NC→NClu" is denoted as the embedding obtained after training on node classification as input for clustering) and adopts normalized mutual information (NMI) to assess the quality of the clustering results. We report test accuracy (%) on NC, NMI (%) on NClu, AUC (%) on LP and $R^2$ (%) on NR tasks. The average results and standard deviations of performing 5 different random seeds on each task are reported.

**Condensation setting**. For condensation methods, the hyperparameters are set according to the original paper if available. Otherwise, we follow the hyperparameter combinations in GC-Bench [42]. Except for the OGB-Arxiv dataset which uses the normalized features provided by [60], all other graph data are condensed using the original features. For our proposed PreGC, we set the importance coefficient $\gamma = 0.5$ in Eq. (8) by default, $\delta t_{\min} = 0.1$ for all datasets.

**Evaluation setting**. To verify the effectiveness of various graph condensation strategies, all graphs, except H&M adopt one-layer GNN encoder, are uniformly configured with a two-layer SGC as the GNN encoder and a single-layer MLP as the predictor by default to adapt to different downstream tasks. For all GNNs, the hidden unit is set to 64, with a learning rate of 0.01, weight decay of 0.0005 and the Adam optimizer employed.

**Experiment environment**. The experiments are conducted on the machine with Intel(R) Xeon(R) Silver 4314 CPU @ 2.40GHz, and NVIDIA GeForce RTX 4090 with 24GB memory and CUDA 12.0. The operating system is Ubuntu 20.04.6 with 384GB memory.

## D More Experimental Results

### D.1 Performance of PreGC on OGB-Arxiv and H&M datasets.

In this section, we show the experimental results at other condensation ratios, as shown in Table E1. For transfer tasks such as "Y→T", the condensed graph is first generated on the source task "Y". The testing phase involves: (1) computing the transport plan between original and condensed graphs, then (2) transferring the labels of task "T" from the original graph to the condensed graph for evaluation.

In particular, existing condensation methods are incompatible with node regression tasks, making condensed graphs unavailable when the source task is "P". Not surprisingly, PreGC consistently outperforms existing GC methods. In particular, the fine-tuning PreGC,i.e., PreGC$_{ft}$, achieves more significant gains on the transfer task. This suggests that the semantic distribution will be significantly different for different tasks. However, existing methods can only be adapted to a single categorization task, resulting in limited application of their condensed graphs.

Table E1: Performance comparison to baselines on different supervised tasks.

| | OGB-Arxiv ($r = 2.5\%$) | | | | H&M ($r = 5.0\%$) | | | |
|---|---|---|---|---|---|---|---|---|
| | T→T | Y→T | Y→Y | T→Y | C→C | P→C | P→P | C→P |
| Whole | 64.43 ± 0.24 | | 55.37 ± 0.21 | | 77.07 ± 0.26 | | 50.80 ± 0.64 | |
| GCDM | 35.49±3.01 | 44.59±1.59 | 45.69±0.95 | 34.66±3.24 | 59.69±1.18 | N/A | N/A | 19.23±1.87 |
| Gcond | 58.49±0.63 | 49.92±1.66 | 48.88±1.05 | 47.18±1.06 | 51.94±2.71 | N/A | N/A | 15.35±1.46 |
| SFGC | 59.72±0.26 | 47.93±0.57 | 50.06±0.56 | 45.66±1.06 | 59.31±2.28 | N/A | N/A | 14.94±0.55 |
| SGDD | 60.46±0.33 | 51.84±0.40 | 49.89±0.53 | 44.50±1.64 | 62.44±0.51 | N/A | N/A | 17.75±3.19 |
| GDEM | 56.69±0.70 | 34.19±0.03 | 49.94±0.63 | 48.54±0.62 | 51.71±0.89 | N/A | N/A | 11.73±1.72 |
| CGC | 59.71±0.42 | 56.78±1.23 | 50.79±0.29 | 49.71±0.54 | 66.60±0.29 | N/A | N/A | 29.79±0.64 |
| **PreGC** | **62.66±0.47** | **61.99±0.49** | **52.96±0.35** | **52.05±0.42** | **71.06±0.26** | **69.38±0.41** | **36.94±1.05** | **36.23±0.87** |
| **Imp.** | 2.20↑ | 5.21↑ | 2.17↑ | 2.34↑ | 4.46↑ | - | - | 6.44↑ |
| **PreGC$_{ft}$** | **62.90±0.49** | **62.85±0.48** | **53.11±0.17** | **53.20±0.21** | **71.54±0.40** | **71.44±0.32** | **37.07±0.96** | **37.01±0.94** |
| **Imp.** | 2.44↑ | 6.07↑ | 2.32↑ | 3.49↑ | 4.94↑ | - | - | 7.22↑ |

## D.2 Generalization Ability of PreGC across Different GNNs.

This section presents additional generalization results of PreGC across different GNNs, as detailed in Tables E2-E4. Overall, the condensed graphs synthesized via PreGC consistently outperform other condensation methods across diverse datasets, demonstrating the effectiveness of graph diffusion augmentation. While the most competitive baseline CGC surpasses PreGC on certain GNNs, its performance remains limited to specific architectures and shows inferior results on some spatial GNNs (e.g., $k$-GNN and GraphSAGE).

Table E2: Node classification performance across different GNNs on Cora ($r = 2.6\%$) ( Highlight marks the lossless results and **Avg.** is the average results of GNNs).

| | SGC | GCN | APPNP | $k$-GNN | GAT | SAGE | SSGC | Bern. | GPR. | **Avg.** |
|---|---|---|---|---|---|---|---|---|---|---|
| Whole | 80.8±0.8 | 81.0±0.6 | 81.9±0.8 | 77.7±1.0 | 81.8±0.8 | 79.4±0.5 | 81.3±0.6 | 81.5±0.3 | 81.2±0.8 | 80.7±1.4 |
| GCDM | 69.2±0.4 | 67.4±0.9 | 74.3±0.3 | 44.8±2.3 | 68.2±0.8 | 46.6±1.1 | 73.9±0.7 | 70.4±1.2 | 66.4±2.5 | 64.6±11.1 |
| GCond | 79.0±0.6 | 79.1±0.5 | 78.7±0.8 | 63.2±3.1 | 73.3±1.6 | 66.7±1.5 | 78.5±1.1 | 74.4±3.0 | 72.0±1.6 | 73.9±5.8 |
| SFGC | 79.2±1.4 | 80.0±0.5 | 80.3±0.5 | 78.2±0.7 | 79.9±0.7 | 77.5±0.6 | 80.7±0.7 | 80.0±0.7 | 79.6±0.4 | 79.5±1.0 |
| SGDD | 81.1±0.8 | 79.8±0.7 | 81.2±1.4 | 71.3±2.6 | 79.2±0.6 | 77.4±0.9 | 80.2±1.0 | 79.5±2.4 | 78.8±1.4 | 78.7±3.0 |
| GDEM | 80.8±0.4 | 80.8±0.4 | 81.1±0.9 | 78.6±0.7 | 80.3±0.4 | 79.1±0.4 | 81.2±0.3 | 80.9±0.5 | 80.6±0.4 | 80.4±0.9 |
| CGC | **81.8±0.6** | 81.5±0.9 | 81.6±0.8 | 40.4±7.0 | **81.0±0.9** | 66.4±1.2 | 81.1±0.3 | **81.7±0.6** | 81.4±0.6 | 75.2±14.0 |
| **PreGC** | 81.6±0.9 | **81.84±1.1** | **82.3±0.4** | **78.8±1.0** | 80.0±1.0 | **79.8±0.6** | **82.1±1.0** | 81.1±0.5 | **82.3±0.6** | **81.1±1.3** |

Table E3: Node classification performance across different GNNs on Citeseer ($r = 0.9\%$).

| | SGC | GCN | APPNP | $k$-GNN | GAT | SAGE | SSGC | Bern. | GPR. | **Avg.** |
|---|---|---|---|---|---|---|---|---|---|---|
| Whole | 69.2±0.4 | 69.6±0.33 | 69.9±0.7 | 65.5±1.3 | 69.5±0.7 | 70.0±0.6 | 69.8±0.7 | 68.2±1.0 | 69.4±0.9 | 69.0±1.4 |
| GCDM | 54.7±1.2 | 55.1±0.9 | 56.4±1.6 | 33.7±1.2 | 56.1±1.2 | 41.7±1.2 | 55.9±1.5 | 51.9±0.7 | 49.2±1.9 | 50.5±7.9 |
| GCond | 68.9±1.0 | 70.1±0.7 | 60.1±1.3 | 52.0±3.4 | 49.4±5.9 | 62.7±1.1 | 60.7±2.0 | 59.9±4.0 | 48.8±5.8 | 59.2±7.8 |
| SFGC | 67.1±0.9 | 69.4±0.6 | 65.7±2.4 | 66.0±2.2 | 65.4±1.6 | 65.7±0.7 | 68.4±0.9 | 63.2±2.2 | 69.0±0.8 | 66.6±2.0 |
| SGDD | 68.9±1.6 | 69.1±1.4 | 70.6±1.0 | 69.0±1.9 | 62.1±4.2 | **70.2±0.6** | 69.9±1.2 | 65.3±3.0 | 64.0±2.9 | 67.7±3.1 |
| GDEM | 70.6±0.3 | 69.2±0.8 | 70.1±0.6 | 68.6±0.7 | 69.2±0.7 | 68.4±0.7 | 69.5±0.3 | 69.2±0.8 | 69.5±0.4 | 69.4±0.7 |
| CGC | 70.5±0.4 | **72.4±0.5** | **71.5±0.7** | 63.3±2.0 | 70.4±0.8 | 63.0±1.3 | **72.2±0.8** | **72.2±0.5** | 64.9±1.0 | 68.9±4.0 |
| **PreGC** | **70.6±0.5** | 69.3±1.5 | 69.9±0.7 | **69.1±0.5** | **71.0±1.0** | 69.1±0.7 | 69.9±0.4 | 69.5±0.7 | **69.9±1.3** | **69.8±0.7** |

## D.3 Performance Gap between Condensed Graph and Original Graph

Figs. E1 and E2 show the labeled reconstruction error of different condensation methods. It can be clearly observed that PreGC consistently and significantly outperforms existing graph condensation methods, particularly on the CiteSeer and PubMed datasets. This result not only confirms the effectiveness of PreGC in preserving the critical properties of original graphs during condensation, but also reveals its robustness to architecture dependencies.

Table E4: Node classification performance across different GNNs on OGB-Arxiv-topic ($r = 2.5\%$).

| | SGC | GCN | APPNP | $k$-GNN | GAT | SAGE | SSGC | Bern. | GPR. | **Avg.** |
|---|---|---|---|---|---|---|---|---|---|---|
| Whole | 64.4±0.2 | 63.8±0.5 | 65.4±0.5 | 64.2±0.6 | 63.8±0.3 | 61.8±0.2 | 64.7±0.4 | 63.0±0.2 | 64.5±0.8 | 63.9±1.0 |
| GCDM | 35.5±3.0 | 48.8±1.6 | 48.9±1.5 | 36.1±2.0 | 52.2±1.0 | 35.3±1.7 | 48.4±0.7 | 42.9±3.0 | 49.6±2.3 | 44.2±6.9 |
| GCond | 58.5±0.6 | 58.7±0.6 | 58.1±0.9 | 27.8±2.6 | 39.8±4.6 | 29.2±2.2 | 60.1±0.2 | 52.5±1.4 | 46.8±1.5 | 47.9±12.9 |
| SFGC | 59.7±0.3 | 58.8±1.1 | 59.9±0.8 | 57.0±0.2 | 59.5±0.5 | 50.1±1.3 | 60.4±0.5 | 57.5±1.9 | 58.3±0.8 | 57.9±3.1 |
| SGDD | 60.5±0.3 | 54.7±0.9 | 56.5±0.9 | 44.1±2.8 | 31.1±2.0 | 41.3±1.9 | 60.3±0.4 | 33.9±3.8 | 37.9±3.1 | 46.7±11.5 |
| GDEM | 56.7±0.7 | 54.7±1.5 | 55.3±1.3 | 35.9±3.9 | 52.2±0.8 | 35.0±2.1 | 55.2±1.0 | 48.4±2.4 | 47.0±3.2 | 48.9±8.3 |
| CGC | 59.7±0.4 | 58.7±0.5 | 61.8±0.8 | 31.7±1.1 | 60.5±0.6 | 37.6±1.9 | 60.1±0.2 | 57.1±1.0 | 53.9±0.8 | 53.5±11.0 |
| **PreGC** | **62.9±0.5** | **62.5±0.7** | **64.9±0.3** | **62.9±0.9** | **61.4±0.6** | **59.5±0.5** | **64.2±0.7** | **59.8±1.5** | **63.2±0.5** | **62.4±1.8** |

Table E5: Node classification performance across different GNNs on OGB-Arxiv-year ($r = 2.5\%$).

| | SGC | GCN | APPNP | $k$-GNN | GAT | SAGE | SSGC | Bern. | GPR. | **Avg.** |
|---|---|---|---|---|---|---|---|---|---|---|
| Whole | 55.4±0.2 | 57.7±0.4 | 55.9±0.4 | 56.8±0.5 | 56.1±0.5 | 54.5±0.6 | 55.6±0.4 | 53.9±0.6 | 54.2±0.3 | 55.5±1.3 |
| GCDM | 45.7±1.0 | 50.0±0.1 | 51.0±0.2 | 50.9±0.2 | 50.9±0.2 | 43.7±1.0 | 50.2±0.4 | 41.8±2.1 | 49.9±0.7 | 47.9±3.4 |
| GCond | 48.9±1.1 | 50.8±0.2 | 50.8±0.5 | 31.2±4.6 | 50.9±0.4 | 41.9±0.9 | 51.2±0.2 | 49.6±0.4 | 50.2±0.3 | 47.3±6.7 |
| SFGC | 50.1±0.6 | 52.0±0.5 | 51.3±0.3 | 51.4±0.4 | 51.9±0.5 | 45.4±1.2 | 51.4±0.4 | **51.3±0.1** | 51.6±0.1 | 50.7±2.1 |
| SGDD | 49.9±0.5 | 50.9±0.2 | 50.7±0.1 | 51.1±0.0 | 51.1±0.0 | 42.4±0.1 | 50.6±0.4 | 43.0±3.7 | 48.5±1.3 | 48.7±3.5 |
| GDEM | 49.9±0.6 | 51.2±0.3 | 48.8±0.5 | 50.9±0.2 | 51.1±0.3 | 42.0±1.3 | 51.2±0.2 | 43.0±3.1 | 50.9±0.3 | 48.8±3.7 |
| CGC | 50.8±0.3 | 51.3±0.1 | 51.0±0.5 | 49.1±1.1 | 51.1±0.4 | 39.1±0.8 | 51.2±0.2 | 50.9±0.3 | 50.9±0.1 | 49.5±4.0 |
| **PreGC** | **53.0±0.4** | **53.3±0.3** | **53.5±0.1** | **52.7±0.3** | **53.0±0.2** | **52.1±0.4** | **53.5±0.4** | 51.1±1.0 | **51.9±0.1** | **52.7±0.8** |

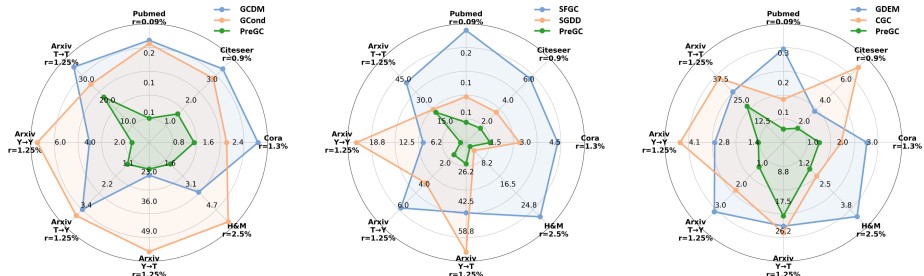

Figure E1: The $LRE$ of different GC methods with the smaller condensation ratio.

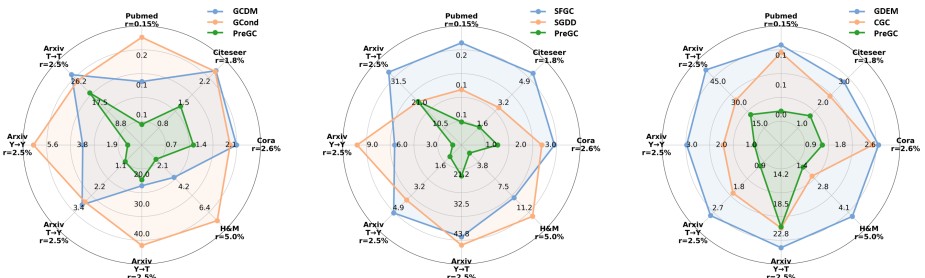

Figure E2: The $LRE$ of different GC methods with the larger condensation ratio.

## D.4 Data Valuation based on Node Significance

This section evaluates the node significance of the original graph, i.e., the ability to explicitly represent the importance of the source node through the transport plan. We rescreened the same number of nodes as the original training set based on Eq. (11). For example, in the Cora dataset, the default number of training nodes is 140 (Default), and we select the top-140 raw nodes based on node significance $\mathbf{s}$ and label them as the training nodes. The training nodes selected through the transport plan prove more valuable than the default training set, yielding significant improvements in the performance of GNN, as shown in Figs. E3 and E4. This matching method not only enhances the effectiveness of data annotation, but also elucidates the relative influence of different nodes during the condensation process, thereby improving the interpretability of graph condensation.

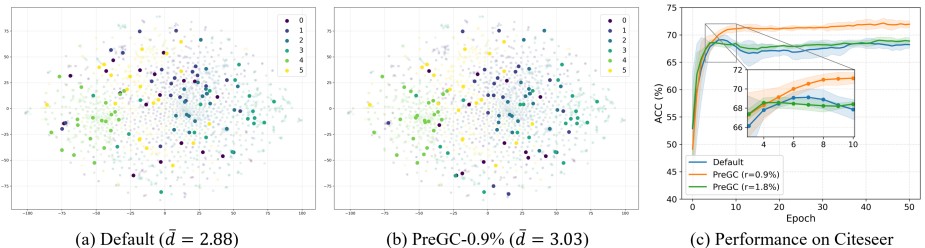

(a) Default ($\bar{d} = 2.88$)  (b) PreGC-0.9% ($\bar{d} = 3.03$)  (c) Performance on Citeseer

Figure E3: Performance of training set selected with transport plan on Citeseer.

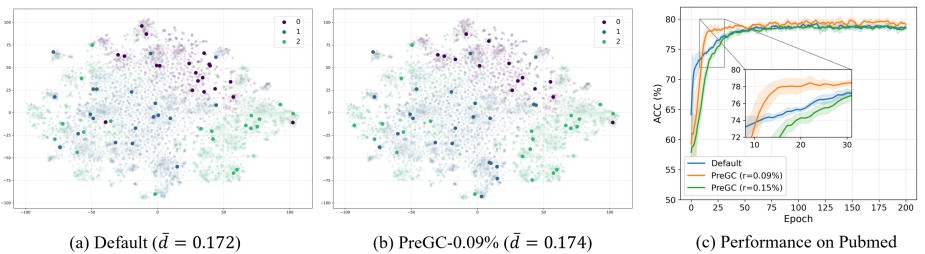

(a) Default ($\bar{d} = 0.172$)  (b) PreGC-0.09% ($\bar{d} = 0.174$)  (c) Performance on Pubmed

Figure E4: Performance of training set selected with transport plan on Pubmed.

## D.5 Sensitivity Analysis

Our investigation further examines the sensitivity of several crucial hyperparameters: the loss balancing coefficient $\xi$ in condensation, the iteration interval $\tau_{up}$ and the decay rate $\epsilon$ in supervised fine-tuning, as illustrated in Fig. E5. The sensitivity to the balance coefficient varies across different datasets. Generally, the best results are achieved when $\xi = 1$ or $2$. Fig. E5 (b) illustrates the impact of supervised fine-tuning on the H&M dataset (The dark blue part represents the condensed graph without supervised fine-tuning), where a trade-off between the values of $\tau_{up}$ and $\epsilon$ is required to ensure optimal performance of PreGC.

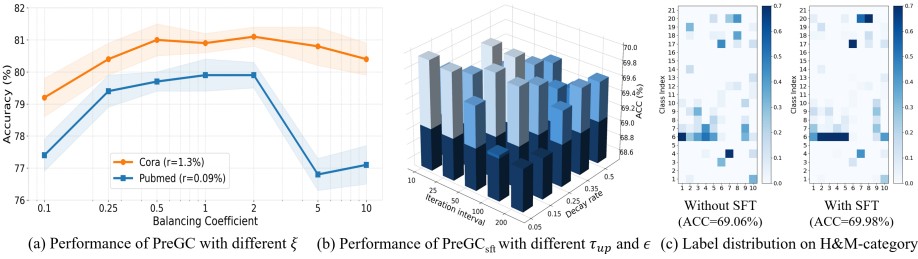

(a) Performance of PreGC with different $\xi$  (b) Performance of PreGC$_{sft}$ with different $\tau_{up}$ and $\epsilon$  (c) Label distribution on H&M-category

Figure E5: The impact of different parameter combinations on PreGC.

## D.6 Ablation Study

In this section, we add ablation experiments of PreGC on the OGB-Arxiv dataset, with the results presented in Table E6. Consistent with the observations on the H&M dataset, both key modules play a significant role in graph condensation. Notably, the removal of graph diffusion augmentation leads to increased performance fluctuations, which demonstrates that graph diffusion augmentation can enhance the generalization capability of the condensed graph.

Table E6: Experimental results for the ablation study on OGB-Arxiv.

| OGB-Arxiv | r=1.25% | | r=2.5% | |
|---|---|---|---|---|
| | Topic | Year | Topic | Year |
| PreGC | 60.55±0.43 | 52.34±0.30 | 62.66±0.47 | 52.96±0.35 |
| PreGC$_{w/o\ Aug}$ | 59.21±0.75 | 51.10±0.64 | 59.93±0.68 | 51.06±0.79 |
| PreGC$_{w/o\ \mathcal{L}_{cost}}$ | 58.84±0.51 | 49.42±0.58 | 59.99±0.42 | 50.55±0.49 |

## D.7 Computational Cost

In this section, we conduct additional experiments to compare the computational efficiency of PreGC with existing baselines on the Cora dataset. It covers peak memory usage and runtime across the pre-processing, condensation, and fine-tuning stages (CGC is excluded as it is a training-free method). The results are shown in Table E7 (* denotes the result of a single execution. Note that in practical implementations, multiple preprocessing often be necessary depending on specific settings.). It can be observed that PreGC achieves significantly faster condensation time compared to most GC methods. Even when fine-tuning is required, it only increases a few additional time costs. Although GDEM spends the least time during the condensation process, it requires extra preprocessing time for eigenvalue decomposition. Overall, both PreGC and GDEM exhibit substantially lower training times than other GC methods. The underlying reason is that existing GC methods employ a nested bi-level optimization: an outer loop updates the GNN parameters, while an inner loop optimizes the condensed graph. In contrast, PreGC leverages graph diffusion, a parameter-free message passing mechanism, eliminating the time-consuming outer loop. This not only reduces condensation time but also prevents the condensed graph from overfitting to architecture-specific parameters. Furthermore, when downstream tasks or label distributions change, existing baselines must re-condense the graph to capture this updated knowledge. In contrast, PreGC is task- and label-agnostic, thus requiring only a single condensation and can be reused multiple times.

Table E7: The computational cost on Cora dataset ($r = 1.3\%$).

| | Memory (MB) | Pre-processing (S) | Condensation (S) | Fine-tuning (S) | Total (S) |
|---|---|---|---|---|---|
| GCDM | 1,276 | - | 1542.38 | - | 1542.38 |
| GCond | 1,280 | - | 2,304.41 | - | 2,304.41 |
| SFGC | 2,514 | 14.01* | 1,923.64 | - | 1,937.65 |
| SGDD | 1,644 | - | 2,364.18 | - | 2,364.18 |
| GDEM | 2,658 | 4.56* | 7.48 | - | 12.04 |
| PreGC | 1,714 | - | 22.36 | 0.81 | 23.17 |

# E Discussion

## E.1 Related Work

**Graph Reduction.** Unlike model quantization [47] and knowledge distillation [37], graph reduction [64, 22] is a data-centric approach. It aims to reduce the size of a graph by decreasing the number of nodes and edges, allowing efficient and effective GNN training. It includes graph sparsification [1, 7], graph coarsening [41, 28], and graph condensation [25, 55]. Graph sparsification aims to reduce edge density while preserving key structural or spectral properties of the original graph. Graph coarsening aggregates nodes or edges to form a smaller super-node or super-edge, often maintaining hierarchical relationships. Their core principle is to maintain specific large-scale graph properties (e.g., spectra and principal eigenvalues) in smaller graphs that have been thinned or coarsened, essentially still

modifying the original graph. In contrast, graph condensation directly optimizes and synthesizes a small-scale condensed graph, which breaks the information constraints imposed by the original graph and promotes consistent GNN test performance.

**Graph Condensation** [25, 24], a.k.a., graph distillation [33], aims to condense large-scale graph data into a small-scale synthetic graph, reducing garph redundancy and accelerating GNN training. Due to its superior ability in graph reduction, it has found a wide range of applications, including accelerating inference [14], graph continuous learning [34], and federated graph learning [51].

Early GC methods inspired by data condensation [61, 62] proposed matching strategies such as gradient matching [25, 24, 13], trajectory matching [64, 59], distribution matching [32, 16, 49], etc., focusing more on learning the consistency of the representation space and proposing a related variant of structure-free. SGDD [55] was the first to propose Laplacian energy distributions to metricize the relationships among structures. GDEM [33] analyzed the upper limit of the existing GC performance is limited by the GNN architecture and improves this problem by eigenbasis matching. In addition, the parameters in different GNNs also bring implicit bias that would be inherited by the condensed graph.[16] and [17] proposed efficient graph condensation without parameters to avoid this problem. [54] synthesized a condensed soft label using multiple self-supervised tasks as auxiliary information to enhance the fitness of condensed graphs. Despite the great efforts made by the above methods, they are still limited to a particular task (e.g., node classification). Recently, SGDC [46] proposed self-supervised graph-level condensation, which is free from task dependency by contrast learning. [15] extended it to self-supervised node-level condensation, improving the generalization ability in condensed graphs on unsupervised tasks. However, the essence of the above methods is still based on class partitioning, which cannot be applied to other supervised tasks (e.g., node regression), hindering the reusability of condensed graphs.

To our best knowledge, we are the first attempt to study generalized graph condensation. PreGC has the advantages of being GNN-parameter-free and task-independent, allowing the condensed graph to be flexibly adapted to arbitrary GNN architectures and diverse downstream tasks.

**Applications of Graph Condensation.** In addition to the aforementioned research, graph condensation has been further extended to other domains [40, 4]. For example, Jin et al. [24] were the first to generalize graph condensation from node-level to graph-level tasks and significantly reduced condensation cost using one-step gradient matching. Xu et al. [50] reformulated the paradigm of graph-level condensation via kernel ridge regression while preserving the performance of the original graph data. Gupta et al. [18] and Wang et al.[46] further extended graph condensation to model-agnostic and self-supervised graph-level settings, rapidly advancing the field. Moreover, owing to the training efficiency of condensed graphs, Liu et al. [35, 36] integrated graph condensation with continual learning, both mitigating catastrophic forgetting and accelerating model training. Yan et al. [51] and Zhang et al. [57] applied graph condensation to federated graph learning, enabling distributed graph learning under privacy constraints by sharing knowledge through condensed graphs.

## E.2 Limitation

Although the proposed PreGC is highly generalizable and reusable, eliminating the need to repeat the condensation, it still suffers from a shortcoming. PreGC inevitably introduces additional complexity to the condensation process due to the computational constraints of optimal transport. Potential improvements in computational efficiency can be explored through graph partitioning techniques or linear optimal transport approximations. However, since the main contribution of this work is to propose a generalized graph condensation paradigm and a pre-trained condensation framework, we defer the study of improving computational efficiency to future research.

## E.3 Broader Impacts

Although graph condensation is pivotal for scaling graph neural networks to large-scale graph data, it can also entail fairness and privacy risks. Due to the condensation process selectively preserves or amplifies structural patterns, it may inadvertently over-represent majority groups while suppressing minority substructures. Moreover, the re-synthesis of node features complicates established anonymization guarantees, as condensed features may be re-identified via topological cues. Accordingly, the condensation procedure should be considered to mitigate unfairness induced by structural bias while simultaneously preventing the leakage of sensitive attribute information.

