# OpenReview forum: "Towards Pre-trained Graph Condensation via Optimal Transport"
_NeurIPS.cc/2025/Conference — NeurIPS 2025 poster_

### Official Review · Reviewer_3v1U · 2025-06-22

**Clarity:** 4
**Significance:** 4
**Originality:** 4
**Rating:** 5
**Confidence:** 4

**Summary:**

This paper proposes a new framework, PreGC, for pre-trained graph condensation that enables task-agnostic and architecture-agnostic usage. While prior methods rely heavily on fixed GNN architectures and supervised labels, PreGC addresses these limitations by leveraging optimal transport theory, graph diffusion augmentation, and transport plan matching. Extensive experiments across tasks, datasets, and GNN architectures demonstrate the effectiveness and flexibility of the proposed approach.

**Questions:**

I like the idea of using graph condensation for graph data valuation. It would be valuable to explore whether other graph condensation methods can also be applied in this context, and how the proposed approach compares to existing graph data valuation techniques.

**Ethical Concerns:**

["NO or VERY MINOR ethics concerns only"]

**Final Justification:**

The authors have addressed my concerns and I am happy to maintain my scores.

**Quality:**

3

**Strengths And Weaknesses:**

Strengths:
1. The paper identifies and addresses critical limitations in existing graph condensation methods, particularly architecture dependency and task supervision.
2. The use of optimal transport and graph diffusion augmentation in the context of graph condensation is novel and theoretically grounded.
3. The experiments involving cross-task generalization are new to this domain and are likely to inspire further research.
4. I also appreciate the exploration of the application of graph condensation to data valuation via node significance, which is an interesting direction that enhances interpretability and practical utility.


Weakness:
1. The paper lacks details on the computational cost of both the pretraining and fine-tuning phases. It is unclear how expensive it is in terms of runtime or resources compared to standard training or existing GC methods. It would be better to report training time, GPU usage, memory footprint, or number of epochs.
2. The description of PreGC’s flexibility in Section 5.2 is not very clear. Specifically, the claim that existing methods are "limited to certain proportions of labels" needs more explanation or supporting examples.
3. The datasets used are relatively small. It would strengthen the work to evaluate on larger or more diverse benchmark datasets, such as those in [1,2].

[1] GC4NC: A Benchmark Framework for Graph Condensation on Node Classification with New Insights. 2024
[2] GC-Bench: An Open and Unified Benchmark for Graph Condensation. 2024

---

> ### Author Rebuttal · Authors · 2025-07-30
>
> We sincerely appreciate your high recognition of our work, particularly regarding the limitations of existing methods and the theoretical foundation of our approach. Furthermore, we are deeply grateful for your endorsement of our proposed node importance-based data evaluation strategy. We will address all your concerns point by point.
>
> **W1:** In response to your recommendations and those of Reviewer **8BSw**, we conduct supplementary experiments to systematically evaluate the computational efficiency of PreGC with existing baselines on Cora dataset. The results are shown in the table below (* denotes the result of a single execution).
> A dedicated section analyzing condensation efficiency will be incorporated in the subsequent version to offer a more thorough demonstration of our research.
>
>
> | r=1.3% | Memory (MB) | Pre-processing (S) | Condensation (S) | Fine-tuning (S) | Total (S)     |
> | ------ | ----------- | ------------------ | ---------------- | --------------- | ------------- |
> | GCDM   | 1,276   | -    | 1542.38    | -    | 1542.38    |
> | GCond  | 1,280    | -    | 2,304.41   | -     | 2,304.41   |
> | SFGC   | 2,514    | 14.01*    |1,923.64   | -     | 1,937.65 |
> | SGDD   | 1,644    | -      | 2,364.18    | -    | 2,364.18   |
> | GDEM   | 2,658    | 4.56*    | 7.48      | -    | 12.04    |
> | PreGC  | 1,714    | -    | 22.36   | 0.81 (optional) | 22.36 / 23.17 |
>
> **W2:**  Incorporating your suggestions and **Reviewer NPxj**’s remarks (C2-3), we conduct additional experiments with other baselines under varying training ratios, as shown in the following Table. Notably, the flexibility of PreGC lies in the fact that it only requires one-time condensation for a given dataset. When the task or label ratio changes, the updated information can be efficiently transferred to the condensed graph by Eq. (14). In contrast, existing methods must recondense the graph to capture new label knowledge, which significantly hinders the reusability of condensed graphs. We will further refine and expand this section in the next version.
>
> | Training Ratio | 0.15    | 0.30 | 0.45  | 0.60    | 0.75  | Condensation times |
> | -------------- | ---------- | ---------- | ---------- | ---------- | ---------- | ------------------ |
> | Whole   | 66.71±0.43 | 69.53±0.11 | 70.44±0.51 | 71.69±0.20 | 71.72±0.57 | - |
> | GCDM  | 35.98±1.31 | 36.32±1.08 | 36.09±0.89 | 36.94±0.60 | 38.35±0.67 | × 5|
> | GCond   | 56.17±0.30 | 56.95±0.31 | 56.83±0.49 | 57.47±0.36 | 58.00±0.34 | × 5|
> | SFGC  | 59.09±0.42 | 59.85±0.38 | 60.04±0.53 | 60.46±0.51 | 60.59±0.48 | × 5 |
> | SGDD | 58.01±0.77 | 58.79±0.40 | 59.03±0.44 | 59.54±0.53 | 60.24±0.46 | × 5  |
> | GDEM  | 52.23±0.32 | 54.20±0.57 | 54.81±0.44 | 54.98±0.48 | 55.41±0.31 | × 5  |
> | CGC  | 58.39±0.33 | 59.04±0.49 | 58.92±0.72 | 59.53±0.56 | 60.05±0.29 | × 5   |
> | **PreGC**  | 60.53±0.32 | 61.96±0.56 | 61.72±0.63 | 63.37±0.49 | 63.58±0.82 | × **1**   |
>
> **W3:**  We appreciate your valuable suggestions. Guided by the comments from both you and the other reviewers (**JS8B** and **BLZo**), we carefully reviewed [1][2] and existing research. We incorporate experiments on two large-scale graphs Reddit and Flickr to systematically assess the generalizability of PreGC. The results are reported in the table below.
>
> | Reddit (r=2.5%) | SGC          | GCN          | APPNP        | k-GNN        | GAT          | SAGE         | SSGC         | Bern.        | GPR.         | Avg.         |
> | --------------- | ------------ | ------------ | ------------ | ------------ | ------------ | ------------ | ------------ | ------------ | ------------ | ------------ |
> | Whole           | 86.1±0.1     | 86.5±0.2     | 85.9±0.2     | 83.7±0.1     | 83.1±0.5     | 83.0±0.2     | 86.8±0.1     | 85.7±0.2     | 84.5±0.3     | 85.0±1.4     |
> | GCDM            | 21.9±1.5     | 24.0±0.6     | 56.3±0.8     | 17.4±3.6     | 15.1±1.4     | 30.6±0.4     | 57.5±0.7     | 53.0±1.4     | 40.9±0.9     | 35.2±16.1    |
> | GCond           | 70.4±1.0     | 69.3±0.5     | 61.8±1.2     | 31.2±0.7     | 23.3±2.6     | 40.8±1.0     | 63.8±0.6     | 64.5±0.5     | 47.2±1.3     | 52.5±16.4    |
> | SFGC            | 65.2±1.0     | 67.7±0.7     | 68.9±3.0     | 33.9±1.3     | 19.6±2.9     | 36.7±2.7     | 59.9±0.3     | 68.9±2.2     | 51.6±2.1     | 52.5±17.2    |
> | SGDD            | 75.5±0.9     | 77.0±0.6     | 68.6±2.2     | 24.6±2.0     | 38.9±2.7     | 46.7±2.8     | 69.6±0.2     | 68.9±1.9     | 51.2±2.2     | 57.9±17.3    |
> | GDEM            | 78.3±0.8     | **79.2±1.0** | 79.1±1.0     | 19.2±2.7     | **77.4±2.4** | 42.4±0.7     | **82.2±0.4** | 69.6±2.2     | 57.2±1.1     | 65.0±20.3    |
> | CGC             | 76.9±0.4     | 78.4±0.3     | 72.5±0.9     | 18.5±6.0     | 50.3±4.3     | 41.3±0.3     | 75.6±0.4     | 75.0±1.1     | 58.8±1.2     | 60.8±19.5    |
> | PGC             | **79.2±0.2** | 74.1±0.4     | **79.9±0.3** | **73.7±0.3** | 72.9±4.1     | **70.3±0.4** | 78.5±0.3     | **75.9±1.3** | **70.3±1.2** | **75.0±3.4** |
>
> | Flickr (r=2.5%) | SGC   | GCN  | APPNP | k-GNN  | GAT  | SAGE | SSGC  | Bern.  | GPR.    | Avg.   |
> | --------------- | ------------ | ------------ | ------------ | ------------ | ------------ | ------------ | ------------ | ------------ | ------------ | ------------ |
> | Whole           | 48.3±0.5     | 47.9±0.1     | 44.4±0.6     | 41.8±2.2     | 43.5±0.9     | 46.9±0.2     | 48.0±0.3     | 46.5±0.3     | 46.2±0.3     | 45.9±2.1     |
> | GCDM            | 46.4±0.8     | 45.5±0.5     | 46.5±0.1     | 42.1±0.1     | 44.9±1.1     | 44.4±0.2     | 45.9±0.3     | 45.4±0.5     | 44.8±0.2     | 45.1±1.3     |
> | GCond           | 47.6±0.2     | 46.5±0.1     | 47.3±0.3     | 43.0±1.7     | 41.9±0.7     | 44.9±1.0     | **47.6±0.1** | **46.3±0.7** | 46.4±0.2     | 45.7±1.9     |
> | SFGC            | 46.0±0.9     | 44.1±0.6     | 45.3±1.6     | 46.8±0.9     | 44.6±1.2     | 44.4±0.2     | 45.8±0.2     | 36.1±6.6     | 45.5±0.2     | 44.3±3.0     |
> | SGDD            | 48.0±0.2     | 43.7±0.9     | 44.7±1.1     | 42.1±0.0     | 30.6±5.9     | 42.1±0.0     | 45.6±0.8     | 42.5±0.7     | 43.6±0.5     | 42.6±4.6     |
> | GDEM            | 42.2±0.0     | 42.1±0.1     | 42.3±0.1     | 42.0±0.1     | 42.4±0.4     | 42.6±0.4     | 42.3±0.2     | 41.5±0.6     | 42.3±0.1     | 42.2±0.3     |
> | CGC             | 47.7±0.5     | **46.9±0.4** | 47.0±0.4     | 47.0±0.4     | 46.8±0.5     | 45.8±0.2     | 46.8±0.3     | 46.0±0.7     | 45.9±0.4     | 46.7±0.6     |
> | PGC             | **48.0±0.4** | 46.5±0.1     | **47.7±0.2** | **47.7±0.7** | **47.4±0.3** | **45.9±0.2** | 47.1±0.2     | 46.0±1.0     | **46.8±0.4** | **47.0±0.7** |
>
>
> **Q1:** Thanks for the positive feedback. Intuitively, leveraging the condensed graph to retrospectively evaluate the node value of the original graph should be a natural idea. This stems from the fact that graph condensation aims to distill the most informative (or valuable) node features and topological structures from the original graph while discarding redundancy or noise. Unfortunately, existing GC methods only implicitly learn certain characteristics of the original graph (e.g., gradients optimized by GNNs, distributions in the representation space, etc.), thereby lacking explicit associations with the original graph. This undoubtedly hinders the transparency or interpretability of the condensation process. PreGC addresses this limitation by introducing a transport plan whose transport probabilities explicitly bridge the condensed and original graphs, providing clearer explanations (i.e., higher transport probabilities indicate greater node contributions). Consequently, existing GC baselines still leave the evaluation of original graph data value an open question.
> As far as we know, current approaches for graph data valuation can be categorized into two types. **(1)** Effective resistance (ER) estimation [3]. This method focuses on quantifying node proximity within graphs and has been widely applied in tasks such as the detection of low-conductance sets in graph clustering [4] and graph spectral sparsification [5]. However, due to its emphasis on pairwise node relationships, ER estimation essentially functions as an edge valuation method rather than a node valuation method. **(2)** Migration of existing data valuation methods (e.g., Shapley, Banzhaf, Datamodels) to GNNs [6]. To adapt to structured data, researchers explicitly incorporate graph structures into the valuation process. For example, when evaluating node $i$, one must first construct an induced subgraph by removing node $i$. Subsequently, a GNN is trained on this induced subgraph to assess the impact of node $i$ on model performance. In short, for a graph with $N$ nodes, this procedure requires generating $N$ distinct $(N−1)\times(N−1)$ subgraphs and performing $N$ complete GNN training evaluations. This renders the approach prohibitively expensive in practice.
> We sincerely appreciate your recognition of our idea and the valuable research direction you have proposed. In future work, we will explore the potential of other GC methods in graph data valuation to further advance the broader applicability of graph condensation.
>
> [1] Gong, Shengbo, et al. GC4NC: A benchmark framework for graph condensation on node classification with new insights. *ArXiv:2406.16715*, 2024.
> [2] Sun, Qingyun, et al. GC-Bench: An open and unified benchmark for graph condensation. NeurIPS, 37, 37900-37927, 2024.
> [3] Lai, Yurui, et al. Efficient topology-aware data augmentation for high-degree graph neural networks. KDD, 1463-1473, 2024.
> [4] Alev, Vedat Levi, et al. Graph Clustering using Effective Resistance. ITCS, 94, 2018.
> [5] Spielman, Daniel A., and Nikhil Srivastava. Graph sparsification by effective resistances. STOC, 563-568, 2008.
> [6] Antonelli, Simone, and Aleksandar Bojchevski. Data Valuation for Graphs. 2025.

---

> > ### Comment · Reviewer_3v1U · 2025-08-02
> >
> > Thanks for the detailed response. I will maintain my positive score (5).

---

> > > ### Author Response · Authors · 2025-08-03
> > >
> > > Dear Reviewer 3v1U, thank you for your positive feedback.  If there are any further clarifications we can provide, please feel free to let us know!

---

### Official Review · Reviewer_JS8B · 2025-06-24

**Clarity:** 3
**Significance:** 3
**Originality:** 3
**Rating:** 4
**Confidence:** 4

**Summary:**

The authors aim to expand the application of graph condensation algorithms to multi-task and multi-architecture scenarios. They first provided a theoretical summary of existing graph condensation methods, demonstrating their shortcomings and proposing a new theoretical framework. Based on the proposed framework, the authors proposed PreGC. They introduced a random diffusion based augmentation to enhance message-passing diversity, and a graph optimal transport based proxy task, which achieved unsupervised pre-training of graph condensation models for adaptability to any model architecture and any downstream task.

**Questions:**

Apart from the weaknesses mentioned above, the proposed method appears to work only on homophily graphs. Does it also work for heterophily graphs?

**Ethical Concerns:**

["NO or VERY MINOR ethics concerns only"]

**Final Justification:**

The theoretical analysis is clear and interesting, and some components of the method are novel. Although the performance in Table 1 is somewhat limited, the proposed method performs well in other settings. Therefore, I maintain my rating of 4.

**Limitations:**

Yes.

**Paper Formatting Concerns:**

NA.

**Quality:**

3

**Strengths And Weaknesses:**

Strengths.

- The authors' theoretical analysis and contributions are concise and excellent. Proposition 3.1 clearly illustrates the limitations of existing work concerning GNN architectures and downstream tasks. Definition 3.1 circumvents these limitations by imposing constraints on representations and graph semantics.
- The idea of introducing random diffusion based augmentation is innovative.
- The experimental setup is interesting, particularly the multi-task configuration for node-level datasets.
- The experimental results on Arxiv and H&M are strong.

Weaknesses.

- The results for cross-task and cross-GNN scenarios on the Cora, CiteSeer, and PubMed datasets are not satisfactory. Other baselines do not consider these scenarios, while PreGC is designed specifically for them. Therefore, I believe the performance improvement of PreGC on these datasets is marginal. The effectiveness of the proposed method may require further clarification.
- The introduction of the methodology is somewhat unclear. Although Appendix B provides some explanation, I feel it is insufficient. The authors should provide more detailed information on how Equations 8 and 9 are solved.
- I suggest the authors provide statistical information table about the datasets.

---

> ### Author Rebuttal · Authors · 2025-07-30
>
> We sincerely appreciate the reviewers' positive evaluation of our work, particularly their recognition of the concise and excellent theoretical analysis and contributions, as well as the interesting and novel experimental design. In what follows, we provide detailed responses to each of the questions raised.
>
> **W1:** Notably, PreGC is not tailored to a particular application scenario. Its primary objective is to distill a task-agnostic and architecture-flexible condensed graph.
> Let us revisit the original motivation behind graph condensation. The original intention of GC lies in accelerating GNN training, which facilitates researchers in developing novel GNN architectures and rapidly testing their performance on certain tasks. However, existing baselines only consider NC task and are limited to specific GNNs (e.g., SGC and GCN). These methods are scenario-constrained and deviate from the original intention of GC.
> In contrast, our method does not focus on specific scenarios but instead follows the general GC paradigm (i.e., Definition 3.1) derived from the perspective of GNN optimization consistency. **(1)** First and foremost, PreGC only requires condensation once on a given dataset and does not need to be re-condensed when tasks or label ratios change. This flexible reusability significantly enhances the practical value of condensed graphs. **(2)** Despite the absence of label signals during condensation, PreGC still outperforms other baselines in most cases on NC task, as shown in Table 1. **(3)** More importantly, the graphs condensed by PreGC consistently outperform existing methods on LP task, further demonstrating that the general GC paradigm enables the condensed graphs to capture and inherit the properties of original graphs. **(4)** Additionally, experiments across different GNNs validate the generalization capability of the condensed graphs, with PreGC achieving the best average performance across nine GNNs.
> Overall, in scenarios specifically tailored to existing GC methods, PreGC still achieves comparable or even sota performance without relying on supervised signals to guide condensation. Moreover, in more generalized scenarios (such as unsupervised tasks, node regression, and cross-task settings), PreGC demonstrates significant superiority over existing baselines. Therefore, the cross-task and cross-architecture evaluations demonstrate the generalizability and reusability of graphs condensed by PreGC, fully aligning with the original intention of graph condensation.
>
> **W2:** We thank the reviewer for this comment. Since Eq. (9) can be regarded as a special form of Eq. (8) (i.e., $\gamma=1$), our discussion primarily focuses on solving for $\pi_\mathcal{D}$ in Eq. (8).  (**PS: Due to the fact that certain mathematical symbols cannot be properly compiled, we have appropriately employed line breaks for clearer representation.**)
> **(1)** Eq. (8) decomposes into two terms: the former is a Wasserstein distance that operates solely on node features. This term can be efficiently approximated using the Sinkhorn-Knopp algorithm [1], which iteratively converges to the optimal transport plan $\pi^* $. Specifically, the Sinkhorn-Knopp algorithm introduces an entropy regularizer and alternates between Sinkhorn projections, $\pi^{(0)}=\exp(-\boldsymbol{\mathcal{K}} (\mathbf{X},\tilde{\mathbf{X}}) / \lambda)$ and $\pi^{(l+1)}=\mathcal{F}_1(\mathcal{F}_2(\pi^{(l)}))$, where $l$ represents the number of iterations and $\lambda$ denotes the weigth of regularization.
> The projections $\mathcal{F}_1(\pi)=\pi\oslash (\mathbf{1}\mathbf{1}^\top \pi)$ and $\mathcal{F}_2(\pi)=\pi \oslash(\pi\mathbf{1}\mathbf{1}^\top)$
>
> denote the column normalization and row normalization, respectively, where $\oslash$ is element-wise division. They ensure that the rows and columns satisfy the given marginal distribution constraints. In the limit, this alternating projection will converge to a minimizer $\lim_{t\to\infty}{\pi^{(t)}}=\pi^* $.
> **(2)** The latter term corresponds to a Gromov-Wasserstein (GW) distance, which can be formulated as a nonconvex quadratic programming problem. We use semi-relaxed Gromov-Wasserstein divergence [2] and optimize $\pi$ with the conditional gradient solver. The gradient of $\pi$ with respect to the GW term is $\nabla_{\pi}=2\boldsymbol{\mathcal{J}} (\mathbf{A},\tilde{\mathbf{A}}) \otimes \pi $. The conditional gradient algorithm consists in solving a linearization $\langle \mathbf{P}, \nabla_{\pi}\rangle $ at each iteration, where $\mathbf{P} \in \mathbb{R}^{N\times M}$ the extreme-point solution that arises from the linearized sub-problem solved. It can be solved by gradient descent with a direction $\mathbf{P}^{(l)} − \pi^{(l)}$, followed by a line search for the optimal step.
> In the optimization process, we utilize the third-party library functions GeomLoss and Python Optimal Transport (POT) to solve Eq. (8) and Eq. (9), respectively. For further details, please refer to [1], [2], and [36]. We will add the above optimization specifics in the next version.
>
> **W3:** We will add statistical tables about the dataset in the next version, including the number of nodes, edges, feature dimensions, edge homophily, and so on.
>
> | Dataset   | Nodes   | Edges      | Features | Classes    | Edge Hom.       |
> | --------- | ------- | ---------- | -------- | ---------- | --------------- |
> | Cora      | 2,708   | 5,429      | 1,433    | 7          | 0.81            |
> | CiteSeer  | 3,327   | 4,732      | 3,703    | 6          | 0.74            |
> | PubMed    | 19,717  | 44,338     | 500      | 3          | 0.80            |
> | OGB-Arxiv | 14,167  | 33,520     | 128      | 40(T) / 5(Y) | 0.70(T) / 0.37(Y) |
> | H&M       | 15,508  | 920,284    | 191      | 21(C)      | 0.34(C)         |
> | Reddit    | 232,965 | 23,213,838 | 602      | 41         | 0.78            |
> | Flickr    | 89,250  | 899,756    | 500      | 7          | 0.32            |
>
> **Q1:**  As a GC method following the generalized graph condensation paradigm, PreGC remains applicable to heterophily graphs. In fact, for the year prediction task on the OGB-Arxiv dataset and the category prediction task on the H&M dataset, they exhibit strong heterophily (edge homophily is 0.37 and 0.34, respectively), and the condensed graph by PreGC still achieves state-of-the-art performance. This is because PreGC learns the spectral properties of the original graph through graph diffusion augmentation, rather than the task-relevant homophily or heterophily characteristics.
>
> Additionally, inspired by feedback from other reviewers (Reviewers **BLZo** and **3v1U**), we have included experiments on the heterophily graph Flickr (edge homophily is 0.32), with the corresponding results available in the following.
>
> | Flickr (r=2.5%) | SGC          | GCN          | APPNP        | k-GNN        | GAT          | SAGE         | SSGC         | Bern.        | GPR.         | Avg.         |
> | --------------- | ------------ | ------------ | ------------ | ------------ | ------------ | ------------ | ------------ | ------------ | ------------ | ------------ |
> | Whole           | 48.3±0.5     | 47.9±0.1     | 44.4±0.6     | 41.8±2.2     | 43.5±0.9     | 46.9±0.2     | 48.0±0.3     | 46.5±0.3     | 46.2±0.3     | 45.9±2.1     |
> | GCDM            | 46.4±0.8     | 45.5±0.5     | 46.5±0.1     | 42.1±0.1     | 44.9±1.1     | 44.4±0.2     | 45.9±0.3     | 45.4±0.5     | 44.8±0.2     | 45.1±1.3     |
> | GCond           | 47.6±0.2     | 46.5±0.1     | 47.3±0.3     | 43.0±1.7     | 41.9±0.7     | 44.9±1.0     | **47.6±0.1** | **46.3±0.7** | 46.4±0.2     | 45.7±1.9     |
> | SFGC            | 46.0±0.9     | 44.1±0.6     | 45.3±1.6     | 46.8±0.9     | 44.6±1.2     | 44.4±0.2     | 45.8±0.2     | 36.1±6.6     | 45.5±0.2     | 44.3±3.0     |
> | SGDD            | 48.0±0.2     | 43.7±0.9     | 44.7±1.1     | 42.1±0.0     | 30.6±5.9     | 42.1±0.0     | 45.6±0.8     | 42.5±0.7     | 43.6±0.5     | 42.6±4.6     |
> | GDEM            | 42.2±0.0     | 42.1±0.1     | 42.3±0.1     | 42.0±0.1     | 42.4±0.4     | 42.6±0.4     | 42.3±0.2     | 41.5±0.6     | 42.3±0.1     | 42.2±0.3     |
> | CGC             | 47.7±0.5     | **46.9±0.4** | 47.0±0.4     | 47.0±0.4     | 46.8±0.5     | 45.8±0.2     | 46.8±0.3     | 46.0±0.7     | 45.9±0.4     | 46.7±0.6     |
> | PGC             | **48.0±0.4** | 46.5±0.1     | **47.7±0.2** | **47.7±0.7** | **47.4±0.3** | **45.9±0.2** | 47.1±0.2     | 46.0±1.0     | **46.8±0.4** | **47.0±0.7** |
>
> [1] Distances, Cuturi M. Sinkhorn. Lightspeed computation of optimal transport. NeurIPS, 26. 2292-2300, 2013.
> [2] Vincent-Cuaz, Cédric, et al. Semi-relaxed Gromov Wasserstein divergence with applications on graphs. ICLR, 1-28, 2022.

---

> > ### Comment · Reviewer_JS8B · 2025-08-04
> >
> > Thanks for the response. My main concerns have been addressed. I would like to maintain my score.

---

> > > ### Author Response · Authors · 2025-08-05
> > >
> > > Dear Reviewer JS8B, we would like to extend our heartfelt gratitude for your active engagement and valuable suggestions.

---

### Official Review · Reviewer_8BSw · 2025-06-30

**Clarity:** 3
**Significance:** 3
**Originality:** 3
**Rating:** 5
**Confidence:** 4

**Summary:**

This paper introduces an innovative pre-training framework PreGC for graph condensation, which overcomes the limitations of existing approaches that are tied to specific tasks and architectures. Utilizing optimal transport, PreGC develops a generalized condensation objective that synchronizes both structural and semantic information between the original graphs and their condensed counterparts. Key contributions encompass a hybrid-interval graph diffusion augmentation technique to enhance architectural generalization, as well as a transport plan matching mechanism to guarantee task-independent semantic consistency. Through comprehensive experiments spanning diverse datasets, tasks, and GNNs, the effectiveness, generalizability, and reusability of PreGC are verified.

**Questions:**

See weaknesses.

**Ethical Concerns:**

["NO or VERY MINOR ethics concerns only"]

**Final Justification:**

Thank you, my concerns have been addressed.

**Limitations:**

Yes

**Quality:**

3

**Strengths And Weaknesses:**

**Strengths**

- The paper re-examines the goal of graph condensation and formulates a generalized optimization framework that integrates existing graph condensation methods under a unified paradigm.

- PreGC is intentionally engineered to be decoupled from both downstream tasks and GNN architectures, rendering it highly reusable and scalable across a wide range of scenarios.

- By incorporating optimal transport, the framework enables semantically meaningful alignment between original and condensed graphs, which has the potential to boost interpretability and traceability.

- PreGC demonstrates consistent superiority over state-of-the-art approaches across various datasets, tasks, and GNN backbones.


**Weaknesses**

- While the paper highlights enhanced performance, it lacks a detailed breakdown of key practical metrics such as training time, memory consumption, and inference latency.

- Despite including ablation study results, these analyses are limited to a single dataset (H&M), which restricts the generalizability of the findings.

- The paper compares PreGC with several recent GC methods but largely overlooks discussions of relevant work in neighboring fields—for instance, data condensation techniques tailored to graph-level tasks [1].

[1] Efficient Graph Continual Learning via Lightweight Graph Neural Tangent Kernels-based Dataset Distillation

---

> ### Author Rebuttal · Authors · 2025-07-30
>
> Thank you for your recognition of this work. Below are our point-by-point responses to your concerns. We believe that the additional experiments and survey will further consolidate and refine our study.
>
> **W1:**  We sincerely appreciate your valuable feedback. In response to your suggestions and those of **Reviewer 3v1U**, we conduct additional experiments to compare the computational efficiency of PreGC with existing baselines on the Cora dataset. It covers peak memory usage and runtime across the pre-processing, condensation, and fine-tuning stages (CGC is excluded as it is a training-free method). The results are shown in the table below (* denotes the result of a single execution. Note that in practical implementations, multiple preprocessing often be necessary depending on specific settings.).
> It can be observed that PreGC achieves significantly faster condensation time compared to most GC methods. Even when fine-tuning is required, it only increases a few additional time costs. Although GDEM spends the least time during the condensation process, it requires extra preprocessing time for eigenvalue decomposition. Overall, both PreGC and GDEM exhibit substantially lower training times than other GC methods. The underlying reason is that existing GC methods employ a nested bi-level optimization: an outer loop updates the GNN parameters, while an inner loop optimizes the condensed graph. In contrast, PreGC leverages graph diffusion, a parameter-free message passing mechanism, eliminating the time-consuming outer loop. This not only reduces condensation time but also prevents the condensed graph from overfitting to architecture-specific parameters.
> Furthermore, as elaborated in our responses to reviewer **NPxj**’s C2-3 and reviewer **3v1U**’s W2, when downstream tasks or label distributions change, existing baselines must re-condense the graph to capture this updated knowledge. In contrast, PreGC is task- and label-agnostic, thus requiring only a single condensation and can be reused multiple times.
> A dedicated section analyzing condensation efficiency will be incorporated in the subsequent version to offer a more thorough demonstration of our research.
>
> | r=1.3% | Memory (MB) | Pre-processing (S) | Condensation (S) | Fine-tuning (S) | Total (S)     |
> | ------ | ----------- | ------------------ | ---------------- | --------------- | ------------- |
> | GCDM   | 1,276       | -                  | 1542.38          | -               | 1542.38       |
> | GCond  | 1,280       | -                  | 2,304.41         | -               | 2,304.41      |
> | SFGC   | 2,514       | 14.01*             | 1,923.64         | -               | 1,937.65      |
> | SGDD   | 1,644       | -                  | 2,364.18         | -               | 2,364.18      |
> | GDEM   | 2,658       | 4.56*              | 7.48             | -               | 12.04         |
> | PreGC  | 1,714       | -                  | 22.36            | 0.81 (optional) | 22.36 / 23.17 |
>
> **W2:** Thanks for your suggestion. We add ablation experiments of PreGC on the OGB-Arxiv dataset, with the results presented in the table below. Consistent with the observations on the H&M dataset, both key modules play a significant role in graph condensation. Notably, the removal of graph diffusion augmentation leads to increased performance fluctuations, which demonstrates that graph diffusion augmentation can enhance the generalization capability of the condensed graph. We will incorporate these results into the manuscript to ensure the generalizability of the findings.
>
> | OGB-Arxiv           | r=1.25%    | r=1.25%    |      | r=2.5%     | r=2.5%     |
> | ------------------- | ---------- | ---------- | ---- | ---------- | ---------- |
> |                     | Topic      | Year       |      | Topic      | Year       |
> | PreGC               | 60.55±0.43 | 52.34±0.30 |      | 62.66±0.47 | 52.96±0.35 |
> | PreGC$_{w/o~Aug}$   | 59.21±0.75 | 51.10±0.64 |      | 59.93±0.68 | 51.06±0.79 |
> | PreGC$_{w/o~Lcost}$ | 58.84±0.51 | 49.42±0.58 |      | 59.99±0.42 | 50.55±0.49 |
>
> **W3:**  We sincerely appreciate your insightful comments. We have noticed that the work [1] you pointed out focuses on the method of graph condensation in specific application scenarios. It innovatively integrated the lightweight graph neural tangent kernel condensation paradigm into graph continual learning, and markedly reduced computational overhead. Motivated by your suggestion, we have systematically surveyed the graph-level GC [1] [2] [3] [4] [5] and the extended applications of GC to additional research directions [1] [6] [10], such as graph continual learning [1] [6] [7] and federated graph learning [8] [9]. **We will devote a dedicated subsection in related work to these developments in the next revision**. We believe that such cross-scenario investigations will enrich the theoretical and methodological landscape of graph condensation and foster sustained advancement in this field.
>
> [1] Qiu, Rihong, et al. Efficient Graph Continual Learning via Lightweight Graph Neural Tangent Kernels-based Dataset Distillation. ICML, 2025.
> [2] Wang, Yuxiang, et al. Self-supervised learning for graph dataset condensation. KDD, 3289-3298, 2024.
> [3] Gupta, Mridul, et al. Mirage: Model-agnostic graph distillation for graph classification. ICLR, 2024.
> [4] Xu, Zhe, et al. Kernel ridge regression-based graph dataset distillation. KDD, 2850-2861, 2023.
> [5] Jin, Wei, et al. Condensing graphs via one-step gradient matching. KDD, 720-730, 2022.
> [6] Liu, Yilun, Ruihong Qiu, and Zi Huang. Cat: Balanced continual graph learning with graph condensation. ICDM, 1157-1162, 2023.
> [7] Liu, Yilun, et al. Puma: Efficient continual graph learning with graph condensation. TKDE, 2024.
> [8] Yan, Bo, et al. Federated graph condensation with information bottleneck principles. AAAI, 39-12, 2025.
> [9] Zhang, Hao, et al. Rethinking Federated Graph Learning: A Data Condensation Perspective. ArXiv:2505.02573, 2025.
> [10] Chen, Dong, et al. Dynamic Graph Condensation. ArXiv:2506.13099, 2025.

---

> ### Author Response · Authors · 2025-08-06
>
> As the deadline approaches, we would like to gently remind the reviewer to check out our rebuttal and join the discussion, as resolving concerns is something we take very seriously, even with positive ratings. We would greatly appreciate it if the reviewer could take a quick look and evaluate whether our current presentation merits an updated rating.

---

> > ### Comment · Reviewer_8BSw · 2025-08-06
> >
> > I would like to thank the authors for their diligent efforts of rebuttal. I am satisfactory with the comprehensive additional results and discussions. Please incorporate these discussions into your final version. Thanks.

---

> > > ### Author Response · Authors · 2025-08-06
> > >
> > > Thank you again for your feedback to help us refine the paper quality. We will revise our manuscript in the final version to highlight the above additional results and discussions.

---

### Official Review · Reviewer_BLZo · 2025-07-02

**Clarity:** 3
**Significance:** 3
**Originality:** 3
**Rating:** 5
**Confidence:** 4

**Summary:**

This paper develops a pre-training graph condensation method to achieve both architecture and task agnostic condensation, and fine-tuning for test-time adaptation, demonstrating the effectiveness both theoretically and experimentally.

**Questions:**

- Typo: line 216 explicittly->explicitly.
- The scalability of the proposed method should be verified on large-scale graphs, such as Reddit, Yelp, and Ogbn-papers100M.

**Ethical Concerns:**

["NO or VERY MINOR ethics concerns only"]

**Final Justification:**

I am satisfied with the thoughtful response of the authors. I decide to keep my positive score.

**Limitations:**

Yes.

**Paper Formatting Concerns:**

No.

**Quality:**

3

**Strengths And Weaknesses:**

Strengths:
- Theoretical insight. The task decomposition of SGC convolution obtains the reconstruction and supervised signal terms, sufficiently illustrating the motivation of the proposed method.
- Comprehensive design. The design of the proposed PreGC involves the parameter-free node representations based on graph diffusion and the semantic alignment to replace the task bias of supervised signals.
- Sufficient experiments. The experimental tasks contain both node and edge tasks, and the compared baselines involve recent state-of-the-art methods.

Weaknesses:
- Validity of assumption. The first proof assumes the existence of an analytic filter $g(\cdot)$, however, it necessitates a discussion on the validity of the assumption.
- Validity of semantic alignment. This paper proposes the semantic alignment to replace the biased supervised signals. Therefore, the gap between semantic alignment and supervised signals requires an in-depth analysis.
- Validity of fine-tuning. The proposed fine-tuning stage at test-time may challenge the fairness of the claimed pre-training technique, compared to other methods.

---

> ### Author Rebuttal · Authors · 2025-07-30
>
> Thank you for your positive comments and recognition of our work, particularly for your high appraisal of the design motivation, methodology, and experimental results of PreGC. In what follows, we will address each of your concerns point by point.
>
> **W1:** We thank the reviewer for pointing out this question. We think that the assumption is both theoretically mild and practically common.
> Firstly, any continuous spectral response can be approximated arbitrarily well by an analytic function on the compact set $\sigma(\mathbf{L})\subset [0,2]$. Therefore, it is reasonable to assume that $g(\cdot)$ is analytic, as it merely rules out pathological, non-smooth filters that are rarely considered in graph learning. Considering a GNN with an analytical filter $g(\cdot)$ is a more common setting [14] [30].
> In addition,  widely used GNN filters satisfy this assumption, such as graph diffusion ($g(\mathbf{L})=e^{-T\mathbf{L}}$) [28], SGC ($g(\mathbf{L})=\mathbf{I}-\mathbf{L}$) [39], and PPNP ($g(\mathbf{L})=\alpha (\mathbf{I}-(1-\alpha )\mathbf{A})^{-1} $) [24], among others. It can more easily generalize the problem to more general scenarios (i.e., Eq. (4)).
>
> **W2:**  Thanks for your profound insights. In fact, we argue that there still exists a discrepancy between semantics and supervisory signals. This is because downstream tasks are manually designed, whereas node semantics are inherent properties of the data itself. To mitigate this discrepancy, we further introduce a fine-tuning strategy, where PreGC_{ft} can be regarded as fitting from semantic alignment to supervisory signal alignment. However, a more intriguing observation is that even without fine-tuning, the graphs condensed by PreGC already achieve sota performance (as shown in Table 2, PreGC). This phenomenon strongly suggests that alignment with the supervisory signal can be effectively approximated by semantic alignment alone. Additionally, in the experiment “Performance Gap between Condensed Graph and Original Graph”, we further validate the differences between various condensed graphs and the original graph via the labeled reconstruction error (LRE). The extremely low LRE of PreGC in Figure 3 indicates that semantic alignment is not only remarkably effective but also superior to existing GC methods that rely on pre-defined labels.
>
> **W3:**  Thank you for your feedback. We ensure that the entire process of PreGC is completely fair. Firstly, PreGC operates without any label guidance during the condensation phase, which guarantees that the condensed graph is aligned with the original graph solely in terms of representation distribution and semantics. However, existing GC methods require assigning labels to each condensed node based on the training set of the original graph before condensation, and explicitly aligning the condensed graph based on its labels. Clearly, this naive approach limits the reusability of the condensed graph in other supervised tasks (especially regression tasks, as shown in Table 2). In contrast, when a graph condensed by PreGC needs to be evaluated on a specific downstream task, it only requires mapping the original training set to the condensed graph by Eq. (14). Even without further fine-tuning, graphs condensed by PreGC still achieve significant performance on most tasks, as demonstrated in Tables 1, 2, and 3 (note that in Tables 1 and 3, we did not employ the fine-tuning at all).
> As you mentioned in the previous question, even for the same graph data, the supervision signals for different tasks may deviate from the semantics. To address this issue, we introduced a fine-tuning strategy (PreGC_{ft}), which is similar to test-time adaptation [1]. Specifically, **(1)** during fine-tuning, PreGC_{ft} adjusts based solely on predicted logits, avoiding any label leakage issues. **(2)** In addition, PreGC_{ft} keeps the condensed graph fixed (i.e., maintaining its structure $\tilde{\mathbf{A}}$ and features $\tilde{\mathbf{X}}$ unchanged) during fine-tuning and only updates the semantic assignment matrix $\mathbf{M}$, making the process highly efficient. **(3)** More importantly, it only requires fine-tuning once for per task.
> Another intriguing finding is that PreGC demonstrates remarkably strong performance even without fine-tuning, as shown in Tables 1 and 3. The improvement of PreGC_ft becomes more pronounced in cross-task transfer settings (such as “Y→T”, “T→Y”, “P→C”, and “C→P” in Table 2). Therefore, we think that in scenarios with a single supervised task requirement, the vanilla PreGC is sufficient, as the graph condensed by PreGC has already effectively captured the essential properties of the original graph with high fidelity.
>
> **Q1:** Thanks for your feedback. The noted issues will be corrected in the next version.
>
> **Q2:** We appreciate your valuable suggestions. Guided by the comments from both you and the other reviewers (**JS8B** and **3v1U**), we have newly incorporated experiments on the large-scale graph Reddit and the heterophilic graph Flickr [1] to systematically assess the generalizability of PreGC. The results are reported in the table below.  It can be clearly observed that on the Reddit dataset, the graph condensed by existing GC methods does not consistently demonstrate ideal performance across each GNN. This is particularly evident in the cases of k-GNN, GAT, and GraphSAGE. In contrast, our proposed method achieves competitive results across all nine representative GNNs and demonstrates compatibility with any GNN architecture, as further evidenced by the average performance metrics.
> Notably, we observe that Yelp [2] is a multi-label graph dataset, which deviates from the typical GC setting (existing baselines must assign a single, unique label to each condensed node before condensation). Additionally, the scale of OGBn-Papers100M is so immense that no baseline has yet reported experiments on it. Owing to these two factors, we are unable to evaluate the performance of existing baselines on these datasets in a short period of time.
> However, it cannot be denied that you have proposed a very innovative setting. Validating the performance of condensed graphs in the multi-label setting would further underscore the value of graph condensation, and we leave this exploration to future work.
>
>
> | Reddit (r=2.5%) | SGC  | GCN | APPNP | k-GNN| GAT  | SAGE    | SSGC  | Bern.   | GPR.   | Avg.  |
> | --------------- | ------------ | ------------ | ------------ | ------------ | ------------ | ------------ | ------------ | ------------ | ------------ | ------------ |
> | Whole           | 86.1±0.1     | 86.5±0.2     | 85.9±0.2     | 83.7±0.1     | 83.1±0.5     | 83.0±0.2     | 86.8±0.1     | 85.7±0.2     | 84.5±0.3     | 85.0±1.4     |
> | GCDM            | 21.9±1.5     | 24.0±0.6     | 56.3±0.8     | 17.4±3.6     | 15.1±1.4     | 30.6±0.4     | 57.5±0.7     | 53.0±1.4     | 40.9±0.9     | 35.2±16.1    |
> | GCond           | 70.4±1.0     | 69.3±0.5     | 61.8±1.2     | 31.2±0.7     | 23.3±2.6     | 40.8±1.0     | 63.8±0.6     | 64.5±0.5     | 47.2±1.3     | 52.5±16.4    |
> | SFGC            | 65.2±1.0     | 67.7±0.7     | 68.9±3.0     | 33.9±1.3     | 19.6±2.9     | 36.7±2.7     | 59.9±0.3     | 68.9±2.2     | 51.6±2.1     | 52.5±17.2    |
> | SGDD            | 75.5±0.9     | 77.0±0.6     | 68.6±2.2     | 24.6±2.0     | 38.9±2.7     | 46.7±2.8     | 69.6±0.2     | 68.9±1.9     | 51.2±2.2     | 57.9±17.3    |
> | GDEM            | 78.3±0.8     | **79.2±1.0** | 79.1±1.0     | 19.2±2.7     | **77.4±2.4** | 42.4±0.7     | **82.2±0.4** | 69.6±2.2     | 57.2±1.1     | 65.0±20.3    |
> | CGC             | 76.9±0.4     | 78.4±0.3     | 72.5±0.9     | 18.5±6.0     | 50.3±4.3     | 41.3±0.3     | 75.6±0.4     | 75.0±1.1     | 58.8±1.2     | 60.8±19.5    |
> | PGC             | **79.2±0.2** | 74.1±0.4     | **79.9±0.3** | **73.7±0.3** | 72.9±4.1     | **70.3±0.4** | 78.5±0.3     | **75.9±1.3** | **70.3±1.2** | **75.0±3.4** |
>
>
> | Flickr (r=2.5%) | SGC          | GCN          | APPNP        | k-GNN        | GAT          | SAGE         | SSGC         | Bern.        | GPR.         | Avg.         |
> | --------------- | ------------ | ------------ | ------------ | ------------ | ------------ | ------------ | ------------ | ------------ | ------------ | ------------ |
> | Whole           | 48.3±0.5     | 47.9±0.1     | 44.4±0.6     | 41.8±2.2     | 43.5±0.9     | 46.9±0.2     | 48.0±0.3     | 46.5±0.3     | 46.2±0.3     | 45.9±2.1     |
> | GCDM            | 46.4±0.8     | 45.5±0.5     | 46.5±0.1     | 42.1±0.1     | 44.9±1.1     | 44.4±0.2     | 45.9±0.3     | 45.4±0.5     | 44.8±0.2     | 45.1±1.3     |
> | GCond           | 47.6±0.2     | 46.5±0.1     | 47.3±0.3     | 43.0±1.7     | 41.9±0.7     | 44.9±1.0     | **47.6±0.1** | **46.3±0.7** | 46.4±0.2     | 45.7±1.9     |
> | SFGC            | 46.0±0.9     | 44.1±0.6     | 45.3±1.6     | 46.8±0.9     | 44.6±1.2     | 44.4±0.2     | 45.8±0.2     | 36.1±6.6     | 45.5±0.2     | 44.3±3.0     |
> | SGDD            | 48.0±0.2     | 43.7±0.9     | 44.7±1.1     | 42.1±0.0     | 30.6±5.9     | 42.1±0.0     | 45.6±0.8     | 42.5±0.7     | 43.6±0.5     | 42.6±4.6     |
> | GDEM            | 42.2±0.0     | 42.1±0.1     | 42.3±0.1     | 42.0±0.1     | 42.4±0.4     | 42.6±0.4     | 42.3±0.2     | 41.5±0.6     | 42.3±0.1     | 42.2±0.3     |
> | CGC             | 47.7±0.5     | **46.9±0.4** | 47.0±0.4     | 47.0±0.4     | 46.8±0.5     | 45.8±0.2     | 46.8±0.3     | 46.0±0.7     | 45.9±0.4     | 46.7±0.6     |
> | PGC             | **48.0±0.4** | 46.5±0.1     | **47.7±0.2** | **47.7±0.7** | **47.4±0.3** | **45.9±0.2** | 47.1±0.2     | 46.0±1.0     | **46.8±0.4** | **47.0±0.7** |
>
> [1] Boudiaf, Malik, et al. Parameter-free online test-time adaptation. CVPR, 8344-8353, 2022.
> [2] Zeng, Hanqing, et al. GraphSAINT: Graph Sampling Based Inductive Learning Method. ICLR, 2020.

---

> > ### Comment · Reviewer_BLZo · 2025-08-03
> >
> > I am satisfied with the thoughtful response of the authors. I decide to keep my positive score.

---

> > > ### Author Response · Authors · 2025-08-03
> > >
> > > We sincerely appreciate your recognition of our work and the time and effort you have devoted as a reviewer!

---

### Official Review · Reviewer_NPxj · 2025-07-03

**Clarity:** 3
**Significance:** 3
**Originality:** 3
**Rating:** 4
**Confidence:** 4

**Summary:**

This submission proposes PreGC, a novel graph condensation framework that distills large-scale graphs into compact synthetic graphs via optimal transport and graph diffusion augmentation. PreGC tries to eliminate task and architecture dependencies in conventional graph condensation methods by (1) using hybrid-interval graph diffusion to enhance generalization across GNN architectures and (2) employing optimal transport plan matching to preserve semantic consistency without task-specific labels. Experiments partially validate PreGC's superiority in cross-task/cross-architecture settings and its interpretability through node significance evaluation.

**Questions:**

Please mainly answer the concerns about the submission listed in C1 and C2.

**Ethical Concerns:**

["NO or VERY MINOR ethics concerns only"]

**Quality:**

3

**Strengths And Weaknesses:**

**Strengths**

S1 Clearly identifies limitations of existing graph condensation methods (task/architecture dependency) and establishes a generalized GC optimization paradigm. Theoretical grounding solidifies the foundation.

S2. Logical flow from problem analysis to solution design, supported by illustrative figures and comprehensive ablation studies.

S3. Evaluated on 5 datasets, 9 GNN architectures, and 4 task types (node/link classification, clustering, regression). Includes sensitivity analysis, interpretability studies.


**Concerns**

C1. Graph diffusion and optimal transport are established techniques. While their integration is novel for GC, the authors are expected to emphasize unique research challenges addressed, e.g., spectral coverage completeness in diffusion for architecture-agnostic GC, transport plan matching for task-agnostic semantic alignment.

C2. This submission needs to focus on strengthening the task-constrained claim verification via more experiments

C2-1: It would be better if authors can expand task diversity to demonstrate broader generalization beyond node/edge-level tasks, including subgraph/graph-level tasks.

C2-2: Based on the pre-trained model with fine-tuning, it is a very classic pipeline to migrate or adapt to different tasks. The basic argument for this pipeline is that the performance of the model trained only on a certain task is not as good as this migration mode. For this reason, the author needs to focus on verifying this point. However, if understand correctly, some results in table 2 are quite conflicting:
(1)	GCDM’s Y→T (T→Y) vs. T→T (Y→Y) suggests GCDM handles some cross-task transfers better, contradicting PreGC’s claims. As shown in Figure 1, the authors argue that existing GC works cannot generalize to other downstreaming tasks.
(2)	PreGC’s Y→T (T→Y) vs. T→T (Y→Y) indicates fine-tuning instability. It is proved that the performance of the preGC model obtained by migration is not as good as that obtained by training only for this task. So, why should we follow the pipeline of “pretrained-finetuned”.

C2-3: Claims like "PreGC uses no task labels" (Line 298) are insufficient. Add experiments: Compare PreGC vs. baselines under limited labeled data (e.g., 10%, 30%, 50% labels) to prove robustness to annotation scarcity.

C3. Please fix some typos, such as “[22, 21, 45] distill...” and [53, 48] replicated”. In addition, the author did not standardize the tense.

---

> ### Author Rebuttal · Authors · 2025-07-30
>
> Thank you for your recognition of our work. In particular, the theoretical analysis and summary of the existing limitations represent one of the most significant contributions in our study. Below, we will address your concerns and questions point by point to clarify any potential confusion.
>
> **W1:** We are deeply grateful for your perceptive and constructive feedback. We fully agree with your perspective. In particular, the spectral coverage completeness guarantees that the condensed graph faithfully preserves the spectral properties of the original graph, while the task-agnostic semantic alignment enables unsupervised graph condensation. To achieve these two pivotal objectives, we devise two elegant mechanisms, i.e., graph-diffusion augmentation and transport-plan matching. This also ensures that PreGC strictly conforms to our formulated generalized GC paradigm (Definition 3.1).
> We appreciate your concise summary of our core contributions. We will incorporate the above content in the next version to further highlight our key innovations.
>
> **W2-1:** We sincerely appreciate your insightful comment. Subgraph-level tasks are indeed pivotal for assessing the generalization capability of node-level graph condensation. This constitutes a novel task and a previously unexplored scenario in GC field. Nevertheless, owing to time constraints, it is non-trivial to modify each GC method to adapt to this new task.
> Moreover, the intrinsic discrepancy between node-level and graph-level graph classification makes directly applying PreGC to graph-level tasks not easily achievable. Whereas node-level GC condensed a single large graph into a small graph, graph-level GC aggregates multiple graphs into a compact set while preserving the size of individual graphs. The former primarily aims to capture inter-sample relationships, whereas the latter emphasizes intra-sample associations. Therefore, graph-level GC bears a closer resemblance to typical dataset condensation.
> In future work, we will endeavor to transplant this study into the subgraph/graph-level tasks to investigate the generalizability of our proposed method.
>
> **W2-2:** Thank you for your insightful comments and careful evaluation of our work. We appreciate the opportunity to clarify the reason for following the pipeline of pretrained-finetuned. As noted in [1], we think that an ideal pre-trained model should satisfy two fundamental criteria. (**C1**) achieving competitive performance across diverse downstream tasks without fine-tuning, and (**C2**) demonstrating further performance improvement after task-specific fine-tuning. These criteria are equally applicable to graph condensation. First, without any fine-tuning, graphs condensed by PreGC already achieve comparable or sota performance across various tasks (as shown in Tables 1, 2, and 3), confirming that PreGC satisfies (**C1**). Second, we can observe from Table 2 that after fine-tuning (PreGC_ft), the condensed graphs exhibit additional performance gains on specific tasks. For example, rising from 60.55 to 60.81 on “T→T” and from 52.34 to 52.47 on “Y→Y” in Table 2, respectively. This demonstrates that PreGC_ft fully aligns with (**C2**).
> Furthermore, regarding the concern about the fine-tuning of PreGC (Q2), this does not indicate instability in PreGC's fine-tuning. On the contrary, the condensed graphs achieve comparable performance after fine-tuning on the same task. For instance, for the OGB-Arxiv dataset in Table 2, the initial performance of the condensed graph directly migrated from the year task to the topic task (“Y→T”) is only 59.62 (PreGC). After fine-tuning, the performance improved to 60.86 (PreGC_{ft}), reaching similar performance (60.81) as fine-tuning directly on the topic task (“T→T”). Similarly, under all cross-task settings (“T→Y”, “P→C”, and“C→P”), PreGC_{ft} ultimately attains performance nearly identical to that achieved by fine-tuning directly on the source task. This demonstrates the effectiveness of the fine-tuning strategy proposed in this work.
> Regarding (Q1), we indeed observe that, in certain cases, GCDM achieves better cross-task performance than its performance on the source task. This phenomenon that contradicts most GC baselines and appears counterintuitive. However, as mentioned in Section 5.2, the lack of explicit correlations leads to insufficient interpretability in existing GC methods. Moreover, since GCDM performs worst in most scenarios, it fails to satisfy (**C1**). Therefore, our focus remains on emphasizing the aforementioned viewpoint, i.e., PreGC is sufficient to achieve optimal performance in most tasks even without fine-tuning. And fine-tuned PreGC_ft can further enhance the quality of the condensed graph, particularly in cross-task settings.
>
> **W2-3:** Incorporating your suggestions and **Reviewer 3v1U**’s remarks (**W2**), we supplement the performance evaluation of various GC methods under different training ratios. The following table demonstrates the performance of condensed graphs on the OGB-Arxiv dataset at the 1.25% condensation ratio. It is noteworthy that, despite the absence of any supervisory signals during the condensation phase, PreGC consistently achieves state-of-the-art performance across varying training ratios. This demonstrates the effectiveness of using semantic alignment as a substitute for supervisory signals. More importantly, existing GC methods require recondensing the graph whenever the training ratio is changed. In contrast, PreGC only requires condensing the original graph once. When the label distribution changes, PreGC can effortlessly transfer the label signals from the original graph to the condensed graph via Eq. (14). This clever strategy demonstrates PreGC's superior flexibility and reusability.
> Thank you again for your constructive feedback. We will incorporate these experimental results into the next version of the manuscript, which will further highlight the unique advantages of PreGC.
>
>
> | Training Ratio | 0.15       | 0.30       | 0.45       | 0.60       | 0.75       | Condensation times |
> | -------------- | ---------- | ---------- | ---------- | ---------- | ---------- | ------------------ |
> | Whole          | 66.71±0.43 | 69.53±0.11 | 70.44±0.51 | 71.69±0.20 | 71.72±0.57 | $~~~-$             |
> | GCDM           | 35.98±1.31 | 36.32±1.08 | 36.09±0.89 | 36.94±0.60 | 38.35±0.67 | $~~\times$ 5       |
> | GCond          | 56.17±0.30 | 56.95±0.31 | 56.83±0.49 | 57.47±0.36 | 58.00±0.34 | $~~\times$ 5       |
> | SFGC           | 59.09±0.42 | 59.85±0.38 | 60.04±0.53 | 60.46±0.51 | 60.59±0.48 | $~~\times$ 5       |
> | SGDD           | 58.01±0.77 | 58.79±0.40 | 59.03±0.44 | 59.54±0.53 | 60.24±0.46 | $~~\times$ 5       |
> | GDEM           | 52.23±0.32 | 54.20±0.57 | 54.81±0.44 | 54.98±0.48 | 55.41±0.31 | $~~\times$ 5       |
> | CGC            | 58.39±0.33 | 59.04±0.49 | 58.92±0.72 | 59.53±0.56 | 60.05±0.29 | $~~\times$ 5       |
> | **PreGC**      | 60.53±0.32 | 61.96±0.56 | 61.72±0.63 | 63.37±0.49 | 63.58±0.82 | $~~\times$ **1**   |
>
> **W3:** Thanks for your feedback. The noted issues will be corrected in the next version. We believe that it will further enhance the readability of this manuscript.
>
> [1] Wang, Zehong, et al. Graph Foundation Models: A Comprehensive Survey. ArXiv:2505.15116, 2025.

---

> ### Author Response · Authors · 2025-08-06
>
> We express our sincere gratitude to the reviewer for dedicating time to reviewing our paper. We have provided comprehensive responses to all the concerns. As the discussion deadline looms within 3 days, we would like to inquire if our responses have adequately addressed your questions. We are more than willing to address any concerns and ensure a comprehensive resolution. Thank you for your time and consideration.

---

> > ### Comment · Reviewer_NPxj · 2025-08-06
> > **Acknowledgement**
> >
> > Thanks for authors' replies. I acknowledge that I have reviewed the author's response and maintain my positive reviews.

---

> > > ### Author Response · Authors · 2025-08-07
> > >
> > > Thank you for your comment! We are glad that our answer addresses your question!

---

### Note · Authors · 2025-08-12

We thank the reviewers' constructive comments and are encouraged that they agree the manuscript presents  **well-justified motivation** (Reviewers NPxj, BLZo, 8BSw, and 3v1U) and **excellent theoretical analysis** (Reviewers NPxj, JS8B, and 3v1U).   The design of the proposed PreGC involves the graph diffusion augmentation and transport plan matching to faithfully **preserve the spectral properties of the original graph** (Reviewers BLZo, 8BSw, JS8B, and 3v1U) and achieve **task-agnostic graph condensation** (Reviewers BLZo, 8BSw, and 3v1U). The reviewers unanimously and highly commended the **sufficiency and novelty** (Reviewers NPxj, BLZo, 8BSw,  JS8B, and 3v1U) of our experiments, as well as the **state-of-the-art performance** of PreGC (Reviewers BLZo, 8BSw, and JS8B).

We sincerely appreciate all reviewers for their unanimous positive feedback and scores on our work. Meanwhile, we are grateful for each reviewer's valuable comments, which we believe will further enhance the quality and impact of this work. Once again, we would like to express our gratitude to all reviewers, AC, SAC, and Pc for their time and effort.

---

### Decision · Program_Chairs · 2025-09-17

**Decision:**

Accept (poster)

**Comment:**

The reviews agree on the technical merits and applicability of PreGC. Therefore, I recommand acceptance.
Key strengths highlighted by reviewers includes "The task decomposition of SGC convolution obtains the reconstruction and supervised signal terms, sufficiently illustrating the motivation of the proposed method." (Reviewer BLZo); "The use of optimal transport and graph diffusion augmentation in the context of graph condensation is novel and theoretically grounded." (Reviewer 3v1U); "PreGC demonstrates consistent superiority over state-of-the-art approaches across various datasets, tasks, and GNN backbones." (Reviewer 8BSw); "The experimental setup is interesting, particularly the multi-task configuration for node-level datasets." (Reviewer JS8B); "Clearly identifies limitations of existing graph condensation methods" (Reviewer NPxj).